# *SOX10* mediates glioblastoma cell-state plasticity

Ka-Hou Man [1,2,8], Yonghe Wu [3,4,8], Zhenjiang Gao[3], Anna-Sophie Spreng[1], Johanna Keding[5], Jasmin Mangei[1], Pavle Boskovic [1], Jan-Philipp Mallm [6], Hai-Kun Liu[3,4,7], Charles D Imbusch [5], Peter Lichter [1] & Bernhard Radlwimmer [1✉]

## Abstract

**Phenotypic plasticity is a cause of glioblastoma therapy failure. We previously showed that suppressing the oligodendrocyte-lineage regulator SOX10 promotes glioblastoma progression. Here, we analyze SOX10-mediated phenotypic plasticity and exploit it for glioblastoma therapy design. We show that low SOX10 expression is linked to neural stem-cell (NSC)-like glioblastoma cell states and is a consequence of temozolomide treatment in animal and cell line models. Single-cell transcriptome profiling of Sox10-KD tumors indicates that Sox10 suppression is sufficient to induce tumor progression to an aggressive NSC/developmental-like phenotype, including a quiescent NSC-like cell population. The quiescent NSC state is induced by temozolomide and Sox10-KD and reduced by Notch pathway inhibition in cell line models. Combination treatment using Notch and HDAC/PI3K inhibitors extends the survival of mice carrying Sox10-KD tumors, validating our experimental therapy approach. In summary, SOX10 suppression mediates glioblastoma progression through NSC/developmental cell-state transition, including the induction of a targetable quiescent NSC state. This work provides a rationale for the design of tumor therapies based on single-cell phenotypic plasticity analysis.**

**Keywords** Glioblastoma; Phenotypic Plasticity; Therapy Resistance; Tumor Cell Quiescence; *SOX10*
**Subject Categories** Cancer; Stem Cells & Regenerative Medicine

## Introduction

Phenotypic plasticity refers to tumor cells' flexible adaptation to external perturbation primarily through non-genetic mechanisms. In the context of cancer treatment, it enables cell-state transition under therapeutic pressure, consequently driving therapy resistance, escape, and ultimately tumor recurrence. It has emerged as one of the driving mechanisms of therapy resistance in many cancers (Chan et al, 2022; Hanahan, 2022; Tsoi et al, 2018).

Glioblastoma, the most common and aggressive adult brain tumor, is a prime example illustrating tumor phenotypic plasticity and its relationship with treatment failure (De Silva et al, 2023; Gimple et al, 2022; Yabo et al, 2022). Despite aggressive multimodal treatment strategies comprising maximal surgical resection and radio-chemotherapy using temozolomide (TMZ), glioblastoma patients typically succumb to the disease within 15 months due to rapid tumor recurrence (Stupp et al, 2005). Treatment failure in glioblastoma is primarily attributed to its extensive phenotypic plasticity that enables tumor cells to withstand the differentiation signals in the brain microenvironment (Brooks et al, 2021), immune surveillance (Gangoso et al, 2021; Liau et al, 2017), and anti-tumor therapy (Liau et al, 2017). Recent glioblastoma RNA sequencing (scRNA-seq) analyses revealed various tumor cell states co-existing along phenotypic gradients spanning states of aberrant neural development, mesenchymal-injury response and metabolic adaptation (Bhaduri et al, 2020; Couturier et al, 2020; Garofano et al, 2021; Neftel et al, 2019; Richards et al, 2021). These plasticity landscapes represent the molecular space in which cancer cells evade therapy by transitioning to resistant cellular states (Chen et al, 2012; Liau et al, 2017; Xie et al, 2022). To develop treatments that block these transitions and direct tumor cells to more manageable states, improving our understanding of the causes and phenotypic consequences of glioblastoma cell-state transitions in disease-relevant models will be essential.

Tumor progression is frequently mediated through co-opting differentiation factors that are critical during normal development (Al-Mayhani et al, 2019; Mu et al, 2017; Suva et al, 2014). Recently, we identified *SOX10* (SRY-Box Transcription Factor 10), which specifies oligodendrocytic lineage differentiation during normal NSC development (Stolt et al, 2002), as a master regulator of the RTK1-subtype of glioblastoma, and demonstrated that *SOX10* suppression promotes tumor progression (Wu et al, 2020). In line with these observations, low *SOX10* expression was reported to be associated with more aggressive and therapy-resistant phenotypes of melanoma (Capparelli et al, 2022; Sun et al, 2014).

Here, we used scRNA-seq analysis to map *SOX10*-dependent cell states in a syngeneic mouse glioblastoma model that reflects essential aspects of therapy-associated human tumor progression, including the emergence of a quiescent stem cell state. Pharmacologic targeting of these stem cells strikingly transformed the cell-state landscape and significantly extended animal survival.

[1]Division of Molecular Genetics, German Cancer Research Center (DKFZ), Heidelberg, Germany. [2]Faculty of Biosciences, Heidelberg University, 69120 Heidelberg, Germany. [3]Shanghai Institute for Advanced Immunochemical Studies, ShanghaiTech University, Shanghai, China. [4]Shanghai Clinical Research and Trial Center, 201210 Shanghai, China. [5]Division of Applied Bioinformatics, German Cancer Research Center (DKFZ), Heidelberg, Germany. [6]Single-Cell Open Lab, German Cancer Research Center (DKFZ), Heidelberg, Germany. [7]Division of Molecular Neurogenetics, German Cancer Research Center (DKFZ), Heidelberg, Germany. [8]These authors contributed equally: Ka-Hou Man, Yonghe Wu. ✉E-mail: b.radlwimmer@dkfz.de

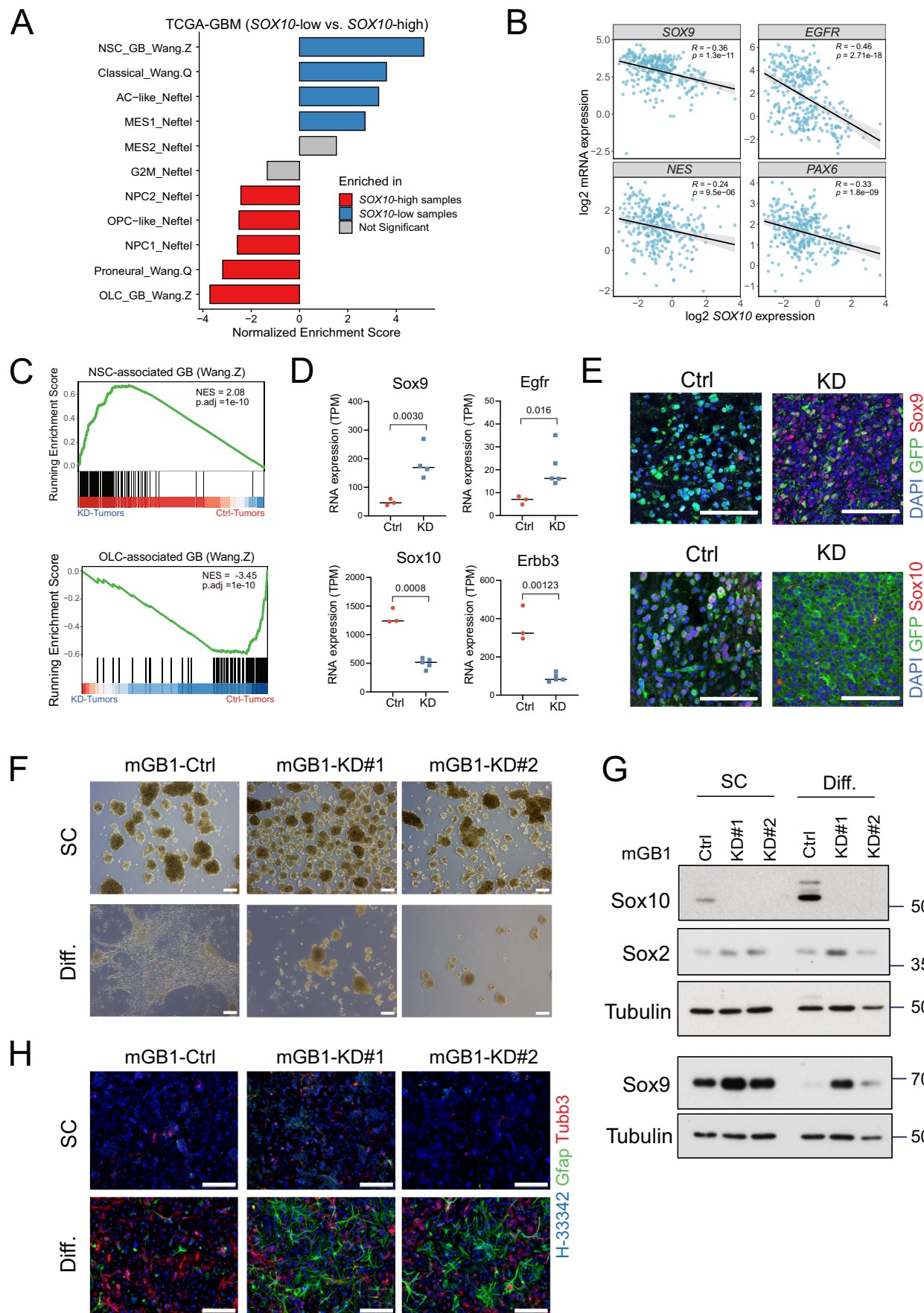

**Figure 1.  SOX10-high and SOX10-low samples occupy distinct glioblastoma cell states.**

(A) Bar plots summarizing the results of gene set enrichment analyses (GSEA) of the indicated genesets curated from several studies. In total, 324 glioblastoma IDH-WT samples of the TCGA cohort were stratified by *SOX10* expression and pre-ranked by log fold change. Differentially expressed genes between the top and bottom 25% of the samples were analyzed by GSEA. (B) Correlation between the expression of *SOX10* with other NSC/AC-like markers in the TCGA-GBM cohort. Pearson correlation tests were used to compute *R* and *P* values. (C) GSEA plots showing enrichment of neural stem cell (NSC)-associated and oligodendrocyte-lineage cell (OLC)-associated glioblastoma gene signatures in mGB1 *Sox10*-KD ($n = 5$) and control ($n = 3$) tumors, respectively. NES Normalized Enrichment Score, *P*.adj adjusted *P* value. Adjusted *P* values (*P*.adj) were calculated using the Benjamini–Hochberg method. (D) RNA Expression (transcripts per kilobase million; TPM) of NSC-associated genes *Sox9* and *Egfr* (top row) and OLC-associated genes *Sox10* and *Erbb3* (bottom row). Mean TPM expression values from control ($n = 3$ animals) and KD ($n = 5$ animals) were shown. Adjusted *P* values between expression levels from mGB1 *Sox10*-KD and control tumors were calculated with the standard 'limma' differential expression analysis. (E) Immunofluorescence images of Sox9 (NSC) and Sox10 (OLC) markers in mGB1 *Sox10*-KD and control mice. Scale bar = 100 μm. (F) Bright-field images of mGB1 control and *Sox10*-KD cells (KD#1 and KD#2) cultured in stem cell (SC) condition (top) or 10% fetal bovine serum (Diff.) for 7 days to induce differentiation (bottom). Scale bars = 200 μm. (G) Western blot analysis of the protein expression of Sox2 in mGB1 control and *Sox10*-KD cells cultured under SC or Diff. conditions. Tubulin was used as the loading control. (H) Immunofluorescence images of Gfap and β3Tubulin (Tubb3) in mGB1 control and Sox10-KD cells (KD#1 and KD#2) seeded in laminin-coated plates under SC (top) or Diff. (bottom) condition. Scale bars = 200 μm. Source data are available online for this figure.

## Results

### SOX10 expression correlates with glioblastoma cell state

Given the role of *SOX10* in the differentiation of neural stem and progenitor cells, we hypothesized that *SOX10* is a key factor determining glioblastoma cell state. Therefore, we analyzed the correlation of *SOX10* expression levels and glioblastoma cell state in 324 primary IDH-WT glioblastoma of the TCGA-GBM dataset. Gene set enrichment analysis (GSEA) using published genesets (Neftel et al, 2019; Wang et al, 2017; Wang et al, 2020) revealed that, consistent with the role of *SOX10* as a pivotal transcription factor of oligodendroglial development, proneural/oligodendrocyte-lineage cell (OLC)-related cellular states were significantly enriched in *SOX10*-high tumors, defined as samples with top quartile expression of *SOX10*. In contrast, neural stem cell (NSC)-related, classical, and astrocyte (AC)-like cell states were enriched in *SOX10*-low tumors (Fig. 1A). In line with these findings, *SOX10* expression was significantly anti-correlated with *SOX9*, *EGFR*, *NES*, and *PAX6* (Fig. 1B), suggesting that *SOX10*-high and *SOX10*-low tumor cells exhibit disparate lineage propensity, with *SOX10*-low tumor cells occupying an AC-like or NSC-like cellular state characterized by high *SOX9* and *EGFR* expression (Cancer Genome Atlas Research, 2008; Neftel et al, 2019; Verhaak et al, 2010).

*SOX10* expression was proposed to be suppressed by therapy-related factors, i.e., radiation treatment in a glioblastoma mouse model (Lau et al, 2015), TGF-β-mediated stress signaling activated by irradiation, hypoxia, or TMZ in human glioma cells (Tabatabai et al, 2006), and BRAF inhibition in melanoma patients (Capparelli et al, 2022; Sun et al, 2014). We confirmed TMZ-induced *SOX10* suppression in the human glioblastoma cell lines cultivated in 2D-adherent culture or in co-culture with iPSC-derived cerebral organoids (Fig. EV1A,B). Western blotting analysis indicated robust expression of the DNA-repair protein O-6-Methylguanine-DNA Methyltransferase (MGMT) in NCH644 but not NCH421k or NCH441 cells (Fig. EV1C), consistent with previously reported TMZ IC50 values: NCH644, 227 mM; NCH421k, 272 μM; (Dirkse et al, 2019). In addition, TMZ-dependent Sox10 downregulation also was observed in a glioblastoma animal model (Rusu et al, 2019) (Fig. EV1D). These findings suggest that the effect of TMZ on SOX10 expression is likely independent of MGMT repair activity and might be mediated by TMZ-induced stress signaling as previously suggested (Tabatabai et al, 2006).

To test whether *SOX10* repression is sufficient to induce OLC-like to NSC-like cell-state transitions, we turned to a syngeneic mouse glioblastoma model, wherein *Sox10*-high mouse glioblastoma (mGB1) neurospheres were transplanted in immunocompetent mice. This model was generated by double knockout in neural progenitor cells and faithfully recapitulates molecular and phenotypic glioblastoma features (Costa et al, 2021). Following intracranial injection of mGB1 control and mGB1 *Sox10* knockdown (*Sox10*-KD) neurospheres, mGB1 *Sox10*-KD tumors developed significantly faster than control tumors, showing features consistent with NSC-like glioblastoma, such as invasive growth at the tumor edges and invasion of the contralateral hemisphere (Fig. EV2A,B) (Wang et al, 2020). RNA-seq data analysis revealed strong enrichment of NSC-like (*P*.adj<0.001) and OLC-like (*P*.adj<0.001) signatures and corresponding marker genes (Wang et al, 2020) in *Sox10*-KD and control tumors, respectively (Fig. 1C,D). The upregulation of the NSC-like glioblastoma marker Sox9 in *Sox10*-KD tumors was confirmed at the protein level by immunohistochemical analysis (Fig. 1E). These data strongly indicate *Sox10* repression as a driver of glioblastoma OLC-like to NSC-like cell-state transition.

To further evaluate this hypothesis, we cultured KD and control cells in serum-free stem cell (SC) and serum-containing differentiating (Diff; 10% FBS) media. Intriguingly, cells appeared unaffected by serum-derived differentiation cues, continuing to grow as free-floating neurospheres, while control cells adopted a more spread-out morphology (Fig. 1F). Western blotting analysis showed that *Sox10*-KD cells upregulated Sox9 and sustained Sox2 expression levels (Fig. 1G). Furthermore, cell-lineage marker analysis by immunofluorescence microscopy revealed the upregulation of the astrocyte-marker Gfap and the essentially unchanged expression of the neuronal marker beta3-tubulin in KD vs. control cells by the supplementation of as little as 2% FBS (Fig. 1H). This observation is consistent with the more NSC/AC-like cell state observed of *Sox10*-low patient tumors (Fig. 1A).

### Loss of SOX10 induces developmental glioblastoma phenotype

To dissect the molecular basis of *Sox10*-KD-mediated cell-state transition, we next performed single-cell RNA sequencing (scRNA-seq) analysis of mouse mGB1 *Sox10*-KD and control syngraft tumors (Fig. 2A). To this end, we intracranially injected GFP-

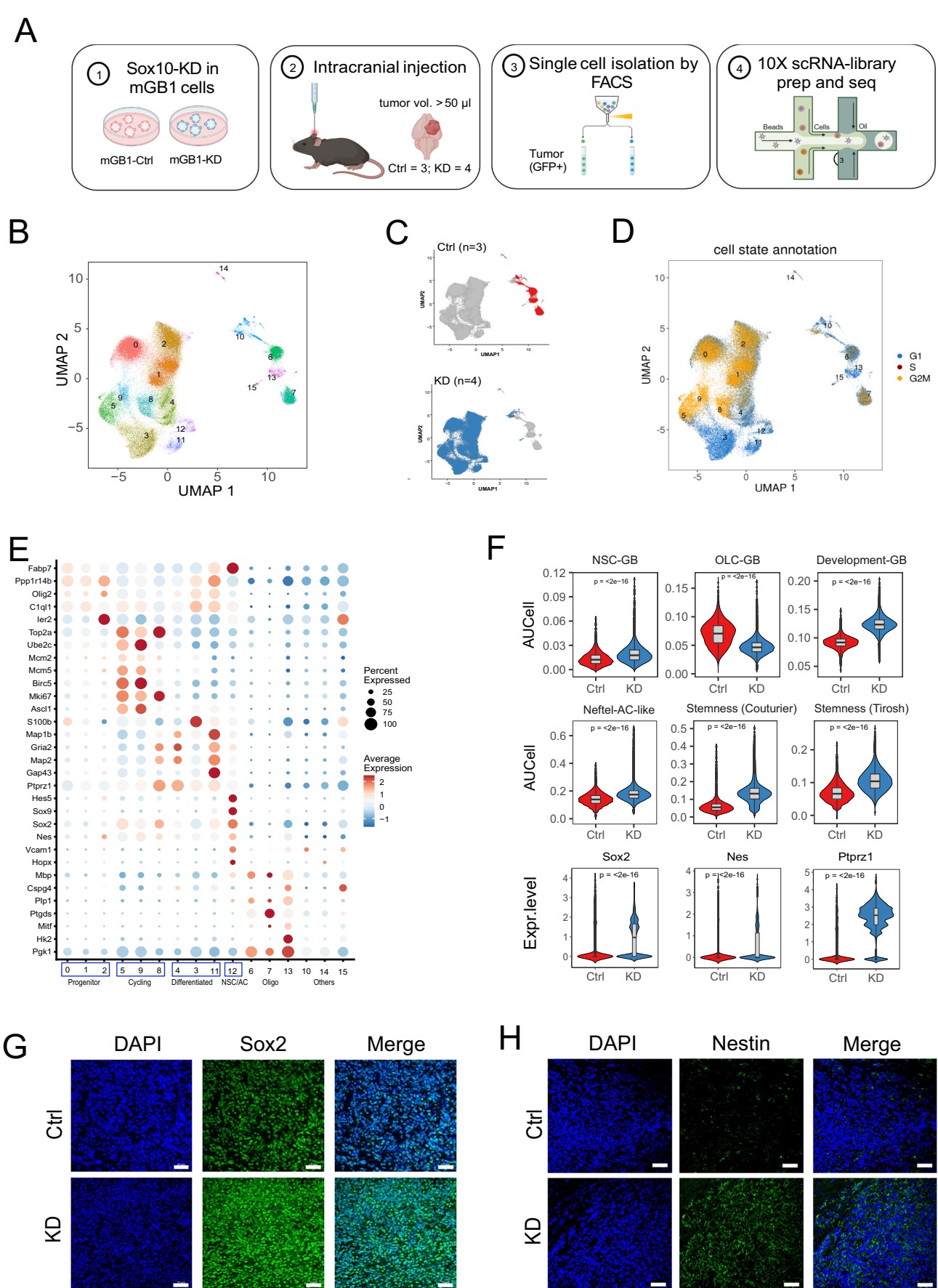

**Figure 2. scRNA-seq analysis reveals divergent cell fates in control and *Sox10*-KD tumors.**

(A) Sample preparation and single-cell sequencing workflow. (B) UMAP visualization of tumor cell transcriptional clusters. (C) UMAP visualization of tumor cell genotypes (top, control; bottom, *Sox10*-KD); red, control cells; blue, *Sox10*-KD cells. (D) UMAP visualization of tumor cell cell cycle phases (G1, blue; S, red; G2/M, yellow). (E) Dot plot summarizing the gene expression of the marker genes (*Y* axis) of each transcriptional cluster (*X* axis) shown in (B). The main clusters in Sox10-KD tumors are indicated by the boxes at the bottom. (F) Violin plots of AUCell scores for selected glioblastoma gene signatures in control and *Sox10*-KD tumor cells: Developmental-like, NSC and OLC-associated, stem cell. Violin plots showing the expression of representative NSC-related genes (*Sox2*, *Nes*, and *Ptprz1*). *P* values were calculated by unpaired two-tailed Wilcoxon's rank-sum tests. $n = 8336$ and 57534 cells in control and KD tumors, respectively. The box-and-whisker plot shows the interquartile range (box), the median AUCell scores or normalized expression values (line inside the box), and the range of data within 1.5 times the interquartile range from the first and third quartiles (whiskers). *P* values were calculated using unpaired two-tailed Wilcoxon's rank-sum tests. (G, H) Validation of the upregulation of Sox2 (G) and Nestin (H) in KD tumors. Scale bars = 50 μm. Source data are available online for this figure.

labeled mGB1 control and *Sox10*-KD cells ($n = 8$, each) into C57BL6/J mice. Upon reaching a tumor volume of 50 μl in T2-MRI imaging, 3 control, and 4 KD tumors were harvested, and tumor cells were sorted based on their GFP expression (Fig. EV2C). A total of 65,870 GFP-positive cells were retained after quality control procedures. We further analyzed the copy number variation profiles of GFP-positive cells and identified large-scale chromosomal aberrations previously described in mGB1 (Costa et al, 2021), confirming that the harvested GFP-positive cells are tumor cells (Fig. EV2D).

Unsupervised hierarchical clustering of GFP-positive cells from control and KD tumors identified 16 transcriptional clusters (Fig. 2B; Dataset EV1). Control and KD cells showed distinct transcription patterns and segregated into two cluster groups (Fig. 2C). Cell cycle scoring analysis indicated a higher proportion of cells in the G2/M cell cycle phase in KD than in Ctrl tumors (44.7% in KD vs. 19.3% in Ctrl). Frequent DNA double-strand breaks were not detected in *Sox10*-KD tumors by γH2AX staining, leading us to conclude that the accumulation of cells in G2/M likely was due to the increased growth rate of KD vs. control tumors rather than DNA-damage-induced cell cycle block (Figs. 2D and EV2E–G).

Differential gene expression analysis identified four major groups of clusters within the KD tumors with characteristics of stages of neural development (Fig. 2E). These included clusters expressing markers of stem/progenitor cells (Arai et al, 2005; Wang et al, 2021; Weng et al, 2019) (clusters 0, 1, and 2), intermediate progenitor, and transit-amplifying cells (5, 8, 9), more differentiated progenies along NSC lineage progression (Raponi et al, 2007) (3, 4, 11). Finally, cluster 12 lacked cell cycle markers but showed high expression of *Sox2* and *Nes* and early astrocytes or quiescent NSCs (qNSC) markers such as *Sox9* and the Notch target *Hes5* (Kalamakis et al, 2019; Llorens-Bobadilla et al, 2015).

Gene signature scoring using AUCell (Aibar et al, 2017) revealed an overall association of KD tumors with NSC development. Specifically, KD tumor cells showed significant ($P < 0.001$) enrichment of NSC-associated-glioblastoma and derichment of OLC-associated-glioblastoma signatures (Weng et al, 2019), and enrichment of AC-like GB (Neftel et al, 2019), developmental glioblastoma (Richards et al, 2021), and cancer stemness gene signatures (Couturier et al, 2020; Tirosh et al, 2016). In addition, they strongly upregulated the NSC markers *Sox2*, *Nes*, and *Ptprz1* (Bhaduri et al, 2020), indicative of a more stem/progenitor-like state associated with tumor aggressiveness (Fig. 2F). The stem- and progenitor-like properties of KD tumors also are evident from the increased Sox2 and Nestin protein expression in KD relative to control tumors (Fig. 2G,H).

Sox2 and Nestin are most prominently expressed in the NSC/AC-like potential qNSC cluster 12 (Fig. 2E). GSEA analysis showed enrichment of two qNSC signatures (Kalamakis et al, 2019), qNSC1 (NES = 2.54; *P*.adj<0.001) and qNSC2 (NES = 2,46; *P*.adj<0.001), further indicating the slow-cycling progenitor characteristics of this cell population (Fig. EV2H). Consistent with these results, cell cycle-scoring analysis assigned G0 status not only to the differentiated-cell clusters 3 and 11, but also to the qNSC cluster 12. In contrast, the "progenitor" clusters 0,1, and 2 showed moderately enriched G2/M signatures (Fig. EV2I). Furthermore, double-staining for the stem and progenitor cell marker Sox2 and the proliferation marker Ki67 in KD tumor sections, Sox2-positive/Ki67-negative cells dispersed throughout the tissue, supporting the presence of a slow-cycling potential qNSC population (Fig. EV2J).

In summary, scRNA-seq analysis indicated the *Sox10*-KD-dependent transition to a more neural development-like tumor phenotype, including fast-cycling proliferative and slow-cycling qNSC cell states that potentially contribute to short-term tumor growth and long-term tumor maintenance. These data suggest that effective therapy needs to target both these cell states.

## *SOX10* suppression enriches a qNSC-like cell state in vitro

We next postulated that *Sox10*-KD leads to comparable cell-state transitions in glioblastoma cell lines, allowing us to use *Sox10*-KD cells as in vitro models to evaluate therapeutic strategies for targeting the quiescent/slow-cycling state. Consistent with this hypothesis, GSEA analysis of mGB1 *Sox10*-KD vs. control RNA-seq data (Wu et al, 2020) showed qNSC (NES = 1.64, adjusted *P* value = 0.0016) and cluster 12 (NES = 1.41, adjusted *P* value = 0.0014) signatures enriched in KD cells (Fig. 3A,B). In addition, qPCR analysis confirmed the upregulation of qNSC-related regulators and markers, including *Sox9* (Llorens-Bobadilla et al, 2015) and *Lrig1* (Marques-Torrejon et al, 2021) (Fig. 3C). Furthermore, *Sox10*-KD increased the abundance of SOX9-positive cells and cells double-positive for SOX9 and the quiescent-cell marker (Coats et al, 1996; Oki et al, 2014) and cell cycle regulator p27 (Fig. 3D,E). We additionally analyzed *Sox10*-KD cells using DNA/Ki67 staining, confirming that KD cells reversibly exited the cell cycle and remained at the G0 phase in reduced growth factor conditions (Fig. EV3A). In line with these findings, we observed p27, SOX9, and *LRIG1* upregulation in human NCH441 and NCH644 *SOX10*-KD stem-like tumor cells (Fig. EV3B–D). Immunofluorescence analysis further revealed increased SOX9 and p27 double-positive cells upon *SOX10* downregulation, accompanied by enhanced tumorigenicity in

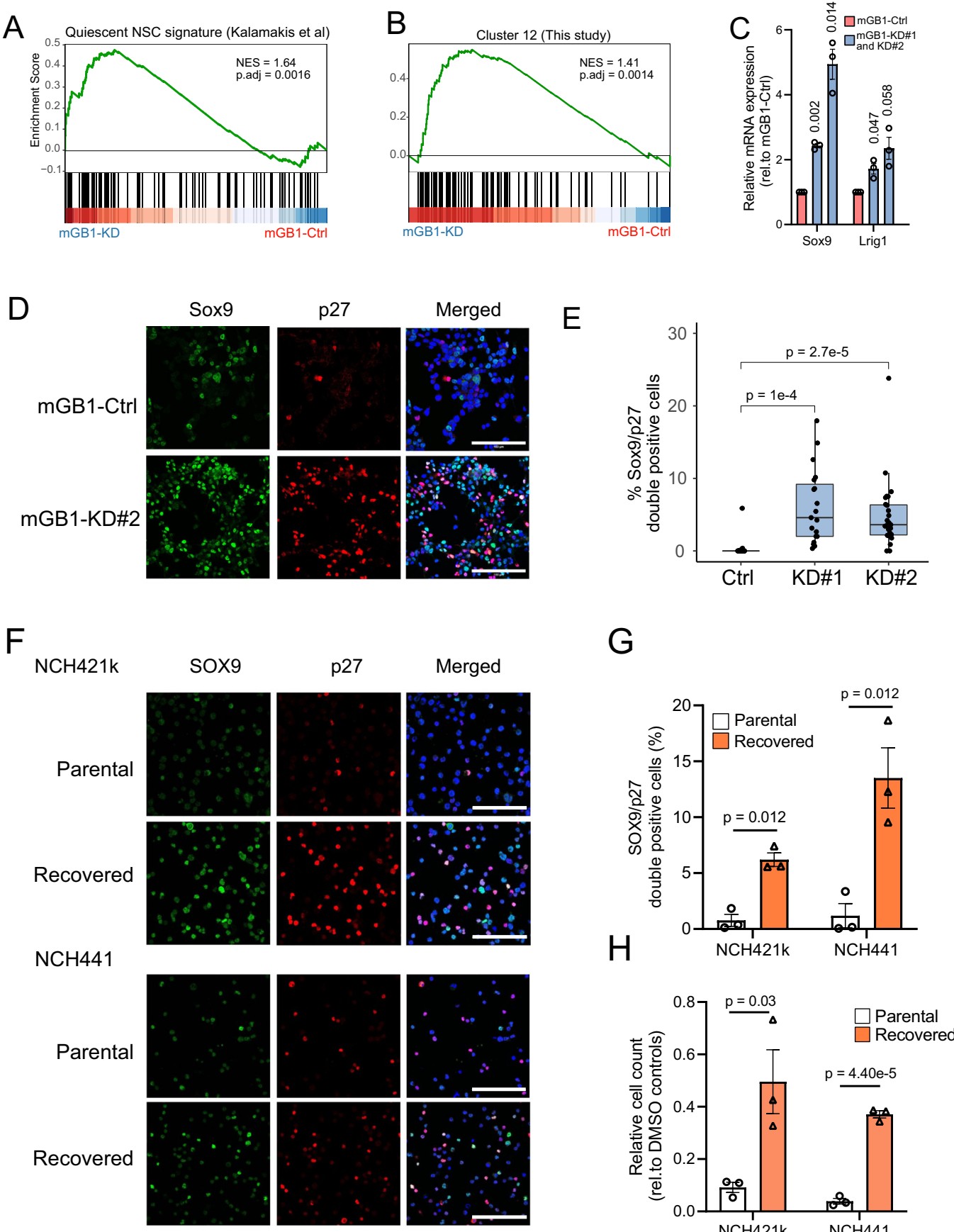

Figure 3. *SOX10 suppression promotes quiescent stem cell-like characteristics.*

(A) GSEA plot showing the enrichment of a qNSC gene signature in mGB1 *Sox10*-KD over control cells. NES, normalized enrichment score; *P*.adj, adjusted *P* values calculated using the Benjamini–Hochberg method. (B) GSEA plot showing the enrichment of Cluster-12 (Fig. 2E) qNSC signature in mGB1 *Sox10*-KD over control cells. The Cluster-12 gene signature comprises the significantly upregulated genes in Cluster 12 with a log2 fold greater than 0.5. Genes were pre-ranked by their log2 expression values in *Sox10*-KD over control cells. NES normalized enrichment scores, *P*.adj adjusted *P* values calculated using the Benjamini–Hochberg method. (C) Quantification of the relative expression of the qNSC markers *Sox9* and *Lrig1* by quantitative real-time PCR. Mean values ± SEM relative to mGB1 control from three independent experiments. *P* values were calculated using one-sample *T* tests. (D) Immunofluorescence co-staining of Sox9 (green) and p27 (red) in mGB1 control and *Sox10*-KD cells under reduced growth factor conditions (1 ng/ml FGF and 1 ng/ml EGF). Blue: DAPI; scale bar: 100 µm. (E) Quantification of Sox9 and p27 double-positive cells. *n* = 19–27 images; The box-and-whisker plot shows the interquartile range (box), the median values (line inside the box), and the range of data within 1.5 times the interquartile range from the first and third quartiles (whiskers). *P* values computed by two-tailed unpaired *T* tests. (F) Immunofluorescence co-staining of SOX9 (green) and p27 (red) in NCH421k and NCH441 parental cells and cells that have recovered from (continued) 100 µM temozolomide treatment. Blue: DAPI; Scale bar = 100 µm. (G) Quantification of SOX9/p27 double-positive cells. Mean ± SEM from three experiments. *P* values were calculated using two-tailed Students' *T* tests. (H) Cell numbers of NCH421k and NCH441 parental and recovered cells. Cell counts were normalized to their respective DMSO controls. Mean ± SEM from three experiments. *P* values were calculated using two-tailed unpaired *t* tests. Source data are available online for this figure.

cerebral organoids (Fig. EV3E). In addition, analysis of a curated brain tumor stem cell transcriptome dataset (Marques et al, 2021) (*n* = 144, 116 independent lines) indicated that *SOX10*-low brain tumor stem cells are enriched for qNSC (Kalamakis et al, 2019) and quiescent stem cell tumor cell (Xie et al, 2022) signature (Fig. EV3F), validating the association of *SOX10*-low and NSC-like state.

Next, we investigated the role of TMZ chemotherapy in the emergence of the p27/SOX9 double-positive state in human glioblastoma cells. To this end, we treated human glioblastoma spheroid cultures with TMZ for 4–5 weeks, after which TMZ was withdrawn and allowed to recover. We observed that the fully recovered cells grew slower than the respective parental cells (Fig. EV3G), were enriched for SOX9/p27 double-positive cells (Fig. 3F,G), and were less sensitive to TMZ than the parental cells (Fig. 3H), suggesting that qNSC-like tumor cells induced by TMZ treatment were maintained in the recovered cells. Immunofluorescence analysis of mouse glioblastoma (Rusu et al, 2019) also confirmed the enrichment of Sox9 and p27-positive cells in TMZ-treated animals (Fig. EV3H).

Collectively, our in vitro analysis indicates that *SOX10* suppression by genetic knockdown or TMZ treatment transitioned human and mouse glioblastoma to a qNSC-like cell state, supporting the use of *SOX10*-KD as an in vitro model for therapy-associated phenotypic transition in glioblastoma.

## Depletion of qNSCs prolongs survival and induces phenotypic plasticity

We next used the *Sox10*-KD cell model to investigate the targeting of the slow-cycling qNSC-like tumor cells. However, since direct-targeting of these cells is typically hampered by their high therapy resistance, we applied a "lock-out" therapeutic strategy that transitions the quiescent state to a faster-cycling state that is more amenable to antiproliferative pharmacologic inhibition (Saito et al, 2010; Takeishi et al, 2013).

Notch pathway activity is crucial in maintaining the slow-cycling stem cell state, particularly in NSC development (Guentchev and McKay, 2006; Imayoshi et al, 2010). Application of the gamma-secretase inhibitor LY411575 at a sublethal concentration of 2 µM resulted in the marked reduction of the NICD and an increase of the cell cycle regulator cyclin D1 at the

protein level in mBG1 cells (Fig. 4A). This was accompanied by a drastic decrease in the expression of the Notch target Hes5 and upregulation of the activated NSC-marker Ascl1 (Fig. 4B), indicating that LY411575 efficiently suppresses Notch pathway activity in *Sox10*-KD cells. Upregulation of Notch target genes was also observed upon *SOX10*-KD in both mouse and human patient-derived tumor spheres (Fig. EV4A). Furthermore, LY411575 significantly reduced the fraction of p27-positive cells in *Sox10*-KD (Fig. 4C,D) but not control cultures (Fig. EV4B,C). In line with this hypothesis, the proportion of 5-Ethynyl-2′-deoxyuridine (Edu)-positive cells was increased by LY411575 treatment (Fig. 4E). These data suggest that Notch inhibition can transition *Sox10*-KD slow-cycling qNSCs toward a faster-cycling cell state.

To test whether LY411575 sensitizes *Sox10*-KD cells to antiproliferative treatment, we performed combination treatments of LY411575 with 78 compounds targeting various proliferation-related signaling pathways, including the ERK and PI3K-mTOR pathways, redox homeostasis and epigenetic regulation (Reagents and Tools Table). Applying a sensitivity index (Chen et al, 2020) threshold of 0.2, we identified several inhibitors with potential LY411575-synergistic effects in KD cells (Fig. 4F). We chose Fimepinostat for further validation, given its favorable safety profile (Younes et al, 2016), wide-ranging inhibitory effect and anti-tumor activity in a preclinical pediatric brain tumor model (Pal et al, 2018; Qian et al, 2012). Pre-treatment of mGB1 *Sox10*-KD cells with LY411575 enhanced their killing by Fimepinostat in the low-nanomolar range (Fig. 4G) and significantly (*P* = 0.031) increased the percentage of apoptotic KD cells (Fig. 4H). Furthermore, analogous to the *Sox10*-KD conditions, a distinct, though statistically not significant, sensitization to Fimepinostat antiproliferative treatment was observed in TMZ-treated human glioblastoma control cells (Fig. EV4D).

Finally, we tested the efficacy of our combination therapy in vivo using our Sox10-KD glioblastoma syngeneic mouse model. Upon most tumors reaching tumor sizes of $2 \times 10^5$ radiation units by bioluminescence imaging (about 50 days post injection), we randomized the mice into either vehicle or a combination of Fimepinostat and LY411575 (Combo) groups (Fig. 5A). Tumor growth was inhibited by the combination treatment (Fig. 5B), resulting in a significant extension of overall survival (*P* = 0.018; log-rank test) compared to the vehicle control (Fig. 5C). Consistent with this finding, histological analyses revealed a significant reduction of proliferating cells in tumors of the

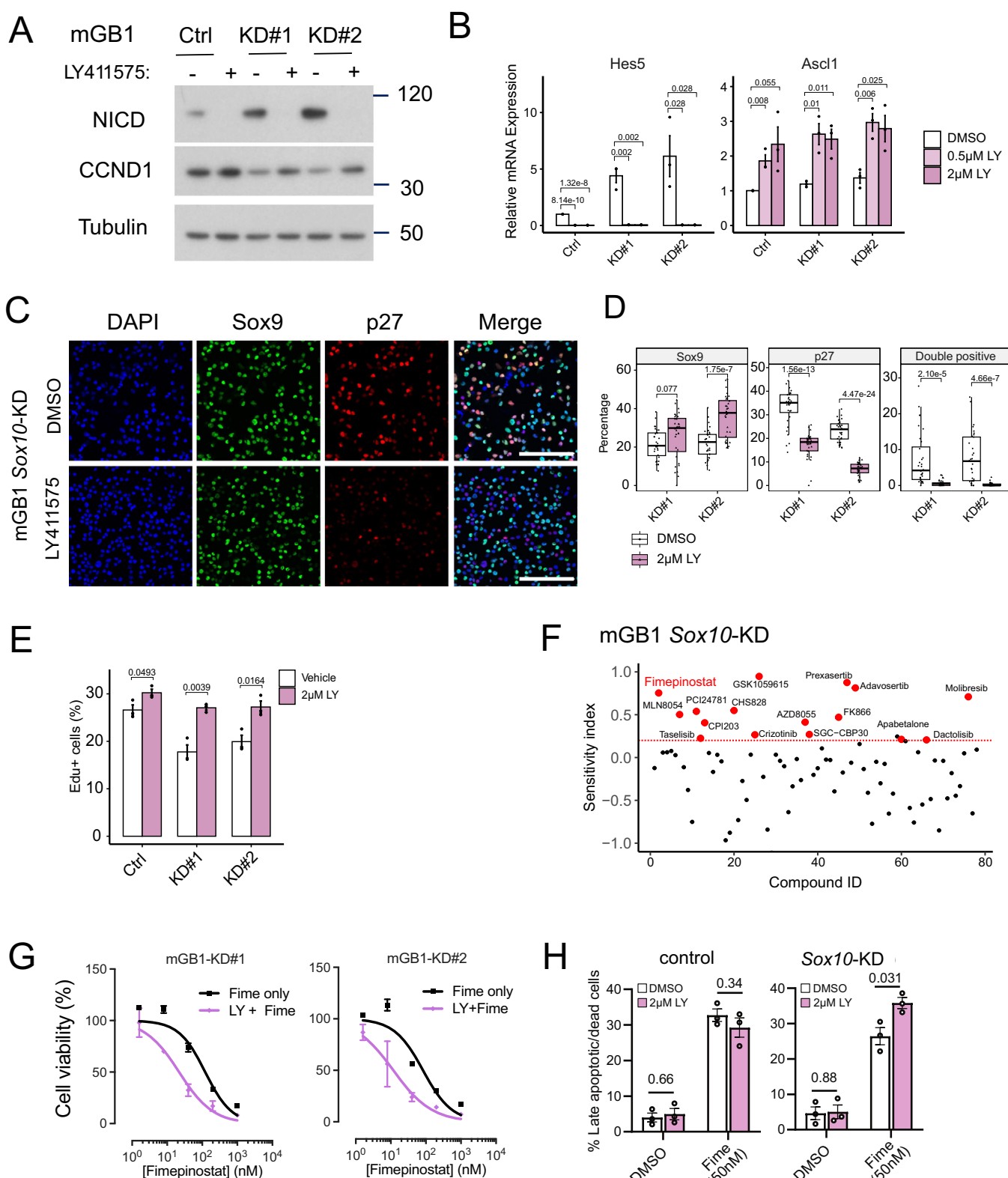

treatment groups (Fig. 5D,E). Fimepinostat alone similarly reduced tumor cell proliferation and extended animal survival, but to a lesser extent than the combination treatment (Fig. EV5).

To characterize cell-state changes associated with treatment responses, we performed scRNA-seq from samples harvested on

day 10 after the initiation of treatment. After removing immune and stromal cells from the dataset, we identified 25,494 tumor cells in 4 samples (*n* = 2 in each group) arranged in 15 transcriptional clusters (Fig. 6A). Following our previous analysis, tumors from the vehicle group showed an overall resemblance to developmental-like

**Figure 4. Notch inhibition sensitizes KD cells to Fimepinostat treatment.**

(A) Western blot analysis of Notch intracellular domain (NICD) fragment and cyclin D1 in mGB1 control and *Sox10*-KD cells treated with 2 μM LY411575 for 48 h. Tubulin was used as a loading control. (B) mRNA expression of the Notch target gene Hes5 upon DMSO or LY411575 treatment at 0.5 μM and 2 μM for 48 h. Shown are mean expression values ± SEM. *n* = 3 biological replicates. *P* values were calculated with two-tailed unpaired *t* tests. (C) Immunofluorescence images of mGB1-KD cells stained with Sox9 and p27 and cultures in reduced growth factor conditions with and without LY411575 treatment. Scale bars = 100 μm. (D) Percentages of Sox9-positive, p27-positive and Sox9 and p27 double-positive cells. The box-and-whisker plot shows the interquartile range (box), the median values (line inside the box), and the range of data within 1.5 times the interquartile range from the first and third quartiles (whiskers). *n* = 32–33 images per condition. *P* values were computed using two-tailed unpaired *T* tests. (E) Bar plots show the percentages of Edu-incorporated cells under each condition. Mean ± SEM from three experiments. *P* values were calculated using a two-tailed unpaired *t* test. (F) Summary of the results of the compound screen in mGB1 *Sox10*-KD (KD#2) cells. The effectiveness of the combination with LY411575 pre-treatment was evaluated by sensitivity index. Compounds that were sensitized by LY411575 treatment (defined as a sensitivity index above 0.2) were labeled red. (G) Dose-response curves of Fimepinostat alone (black curve) and LY411575 + Fimepinostat (purple curve) in mGB1 *Sox10*-KD cells (KD#1 and KD#2). Shown are the mean ± SD of the cell viability (%) from three biological replicates. (H) Bar plots showing the percentages of late-apoptotic cells (Annexin-V and Live/Dead staining double-positive cells) in mGB1 control and *Sox10*-KD cells. Shown are the median values ± SEM. *P* values were computed using two-tailed unpaired *T* tests. Source data are available online for this figure.

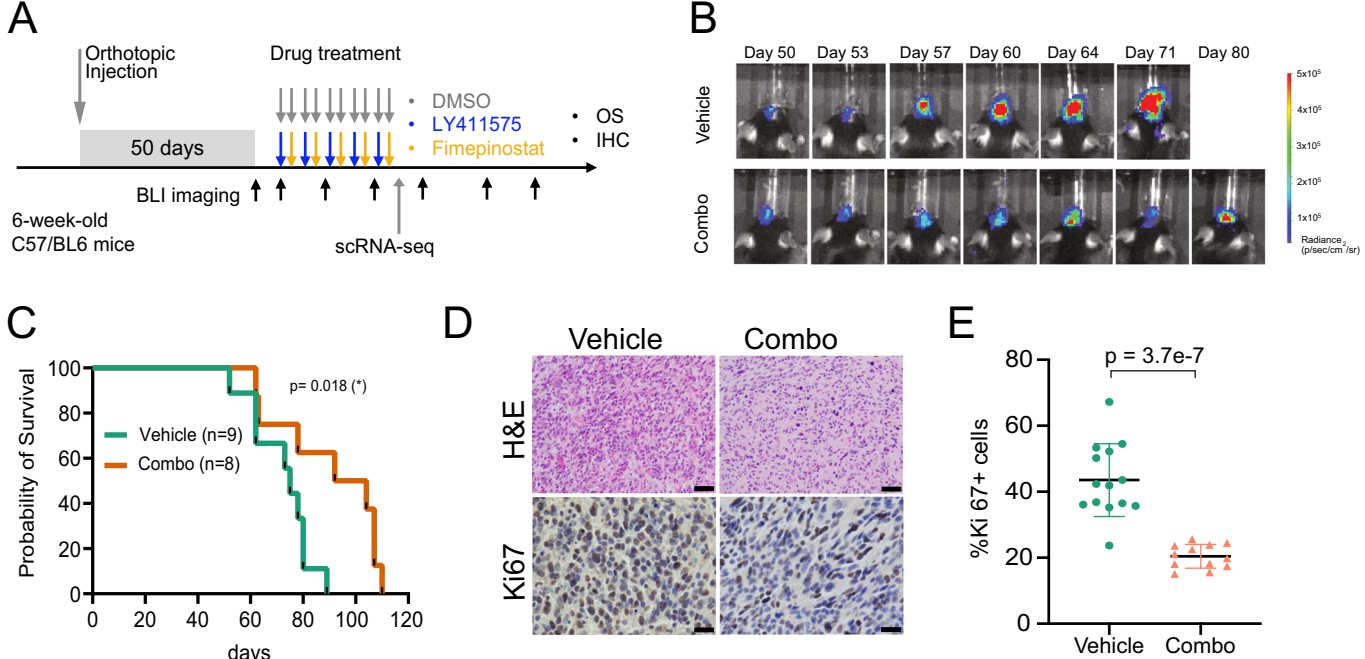

**Figure 5. In vivo validation of combination treatment.**

(A) Schematic summarizing in vivo drug testing experiments. mGB1 *Sox10*-KD cells (KD#2) were orthotopically injected. Upon engraftment and tumor growth, as evidenced by the bioluminescence imaging, animals were randomized into either Vehicle (*n* = 9) or combination (Combo) treatment groups (*n* = 8). In the combination treatment group, the drugs (100 mg/kg Fimepinostat + 10 mg/kg LY411575, p.o.) were delivered on alternating days for five cycles (total of 10 days). (B) Tumor growth visualized by bioluminescence imaging in the Vehicle and Combo groups. (C) Kaplan–Meier survival curves of mice intracranially injected with KD cells in Vehicle (*n* = 9) and Combo groups (*n* = 8), respectively. Log-rank test was used to calculate the *P* value indicated in the plot. (D) H&E and Ki67 staining of end-stage tumors of each treatment group. Scale bars = 50 μm (H&E) and 20 μm (Ki67 IHC). (E) Quantification of Ki67 positive cells in (D). Shown are the median values ± interquartile ranges. *P* values were calculated with two-tailed unpaired *T* tests (*n* = 12–14 images from three different animals in each group). Source data are available online for this figure.

glioblastoma, including qNSC (cluster 11) and proliferative cell populations. UMAP analysis of the individual tumors from the treatment and control groups revealed that the combination treatment depleted the Notch-high clusters 5, 8, and 11 and the proliferative clusters 4 and 5, indicating the specificity of the treatment (Fig. 6B,C). Notably, the combination treatment strategy also resulted in the emergence of new clusters 9, 10, and 13, which were enriched for qNSC and mesenchymal features (Fig. 6D).

Additional analysis of all tumor cells indicated that the combination therapy diverted the tumors from cellular states reminiscent of normal neural development (developmental

glioblastoma) toward inflammatory wound response-related states (injury glioblastoma), along a recently proposed transcriptional gradient (Richards et al, 2021) (Fig. 6E). This observation was further supported by signature scoring of the corresponding developmental and injury genesets (Fig. 6F).

## Discussion

In recent years, numerous scRNA-Seq studies have emphasized the remarkable phenotypic plasticity of glioblastoma cells (Yabo et al,

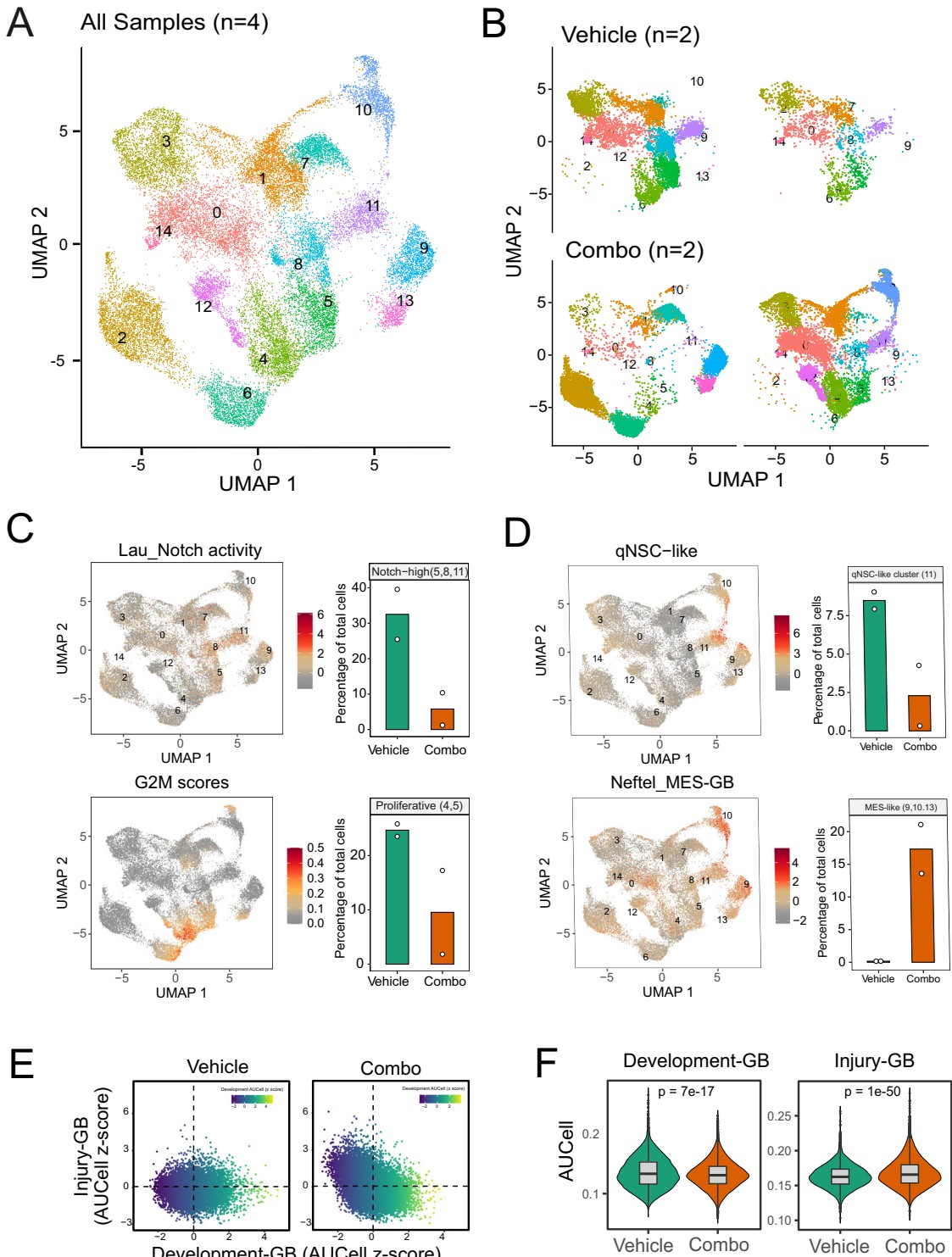

2022) and revealed a phenotypic space defined by neurodevelopmental state and stress adaptation (Garofano et al, 2021; Neftel et al, 2019; Richards et al, 2021). However, the mechanisms and translational consequences of phenotypic plasticity remain largely unexplored. Here, we showed that *SOX10* governs glioblastoma plasticity by mediating the transition between two clinically relevant cell states. Single-cell analyses further revealed the emergence of the quiescent stem cell state upon *Sox10* downregulation, reminiscent of the cell-state transition mediated by TMZ treatment. Importantly, we demonstrated the feasibility and effectiveness of designing a treatment strategy guided by the deep characterization of the tumor cell-state landscape.

A recent study demonstrated that the inactivation of tumor suppressors (Tp53, Nf1, and Pten) in mouse oligodendrocyte-lineage cells (OLCs) versus NSC (NSCs) generated glioblastoma with distinct

**Figure 6. scRNA-seq analysis of post-treatment tumors.**

(A) UMAP visualization of RNA expression clusters from single-cell sequencing analysis of tumors from two animals per treatment group (total $n = 4$). Tumors were harvested on day 5 after treatment started. (B) UMAP visualization of RNA expression clusters of the four individual tumors ($n = 2$ in each group). (C) UMAP plots showing the distribution of cells with high Notch (top) and G2M scores (bottom). Bar plots on the right quantify the depletion of clusters with the highest Notch activity (clusters 5, 8, 11) and proliferative scores (clusters 4 and 5) by combination relative to control treatment. The $Y$ axis shows the proportion of cells in the indicated clusters relative to all cells sequenced for the respective tumor. Shown are the mean values ± interquartile ranges of the percentage of total cells in two animals in each group. (D) UMAP plots showing the distribution of cells with high qNSC-like (top) and MES-GB like scores (bottom). Bar plots on the right quantify the percentages of the cluster with high qNSC-like cell scores (cluster 11) and MES-GB scores (clusters 9, 10, and 13) by combination relative to control treatment. The $Y$ axis shows the proportion of cells in the indicated clusters relative to all cells sequenced for the respective tumor. Shown are the mean values ± interquartile ranges of the percentage of total cells in two animals in each group. (E) Visualization of the dataset using scaled Developmental and Injury Response AUCell scores as $X$ and $Y$ axes, respectively. (F) Violin plot showing the AUCell scores of Development-GB and Injury-GB. $n = 8160$ and 17,334 cells in Vehicle and Combo groups, respectively. The box-and-whisker plot shows the interquartile range (box), the median AUCell scores (line inside the box), and the range of data within 1.5 times the interquartile range from the first and third quartiles (whiskers). $P$ values were calculated using unpaired two-tailed Wilcoxon's rank-sum tests.

phenotypes and therapeutic vulnerabilities (Wang et al, 2020). Here, we showed that the suppression of *SOX10* is sufficient to mediate the transition of OLC-like (*SOX10*-high/*SOX9*-low) to NSC-like (*SOX10*-low/*SOX9*-high) tumor phenotypes, in line with the roles of these transcription factors in determining cell-lineage identities during normal development (Glasgow et al, 2014; Scott et al, 2010; Stolt et al, 2002). Our data further indicate that this phenotypic plasticity is based on the co-option of *SOX10* and *SOX9* regulation in normal development (Reiprich et al, 2017). This antagonistic relationship between *SOX10* and *SOX9* also appears to extend to other cancers, i.e., resulting in melanoma progression upon reduced expression of *SOX10* (Shakhova et al, 2015; Tsoi et al, 2018). This suggests *SOX10* could be a key regulator of phenotypic transition across different cancer types.

Our single-cell sequencing analysis further revealed a heterogeneous cellular architecture of the *Sox10*-KD tumors, characterized by the presence of cycling stem and progenitor cells and a cluster of slow-cycling tumor cells, similar to a type of dormant stem cell that regenerates neural and glial cells after brain injury and in cancer, where quiescent stem cell-like tumor cell populations were associated with tumor initiation and recurrence (Chen et al, 2012; Kalamakis et al, 2019; Llorens-Bobadilla et al, 2015; Xie et al, 2022). Furthermore, we showed that temozolomide treatment suppressed *SOX10* in glioblastoma model systems, resulting in the enrichment of a slow-cycling SOX9/p27 double-positive cell population, suggesting a role for *SOX10*-dependent cell-state transitions in therapy-associated tumor progression. Our results are corroborated by findings suggesting an association between treatment-induced *SOX10* expression and a slow-cycling phenotype in both glioma and melanoma (Capparelli et al, 2022; Cronin et al, 2013; Lau et al, 2015; Sun et al, 2014).

A key question surrounding tumor plasticity is whether we can force therapy-resistant tumor cells to adopt a therapeutically more manageable cell state. The Notch signaling pathway has been implicated in maintaining the slow-cycling NSCs (Boareto et al, 2017; Chapouton et al, 2010; Guentchev and McKay, 2006), glioblastoma stem cells (Liau et al, 2017) and dormant stem cells in other anatomical locations (Aguanno et al, 2019). Here, we showed that *SOX10*-KD leads to decreased Notch pathway activity, potentially through the activation of a positive feedback loop mediated by *SOX9* and *SOX2* (Wang et al, 2019), consistent with our finding of increased protein levels of these factors in KD tumors. We then applied sublethal Notch inhibitor treatment to drive *Sox10*-KD cells out of their slow-cycling phenotype, rendering them sensitive to antiproliferative therapy such as standard radio-chemotherapy (Yahyanejad et al, 2016). On this basis, we hypothesized that Notch inhibitors could also sensitize to the HDAC/PI3K dual inhibitor and suppressor of DNA repair Fimepinostat (Pal et al, 2018). This therapeutic strategy successfully depleted proliferative and slow-cycling qNSC-like tumor cells and significantly extended overall survival in a syngeneic glioblastoma mouse model. However, the applied treatment also markedly re-modeled the tumor-phenotype landscape, leading to the emergence of qNSC-like cell populations with mesenchymal features, representing potential therapy-resistant states. These therapy-induced clusters might be the reason for the modest improvement of survival in the combination treatment group; however, their detailed characterization offers an opportunity to adapt and improve future treatment approaches targeting slow-cycling populations.

In summary, these results show the potential of using phenotypic plasticity information to aid the design and optimization of cancer therapies.

## Methods

**Reagents and tools table**

| Primary antibodies | | | |
|---|---|---|---|
| **Name** | **Dilution** | **Source** | **Catalog number** |
| Anti-Cleaved Notch1 (Val1744) (D3B8) rabbit mAB | 1:1000 (WB) | Cell Signaling Technology | 4147 (RRID:AB_2153348) |
| Anti-Cyclin D1, mouse mAB | 1:1000 (WB) | BD Biosciences | 556470 (RRID: AB_396432) |
| Anti-GFAP antibody, chicken pAB | 1:500 (IF) | Millipore | AB5541 (RRID: AB_177521) |
| Anti-GFP antibody, chicken pAB | 1:500 (IF) | Abcam | ab13970 (RRID: AB_300798) |
| Anti-Ki67, rabbit pAB | 1:200 (FC) | Abcam | ab15580 (RRID:AB_443209) |

| Primary antibodies | | | |
| --- | --- | --- | --- |
| **Name** | **Dilution** | **Source** | **Catalog number** |
| Anti-p27Kip1, mouse mAB | 1:1000 (WB); 1:300 (IF) | BD Biosciences | 554069 (RRID:AB_395225) |
| Anti-SOX2, rabbit pAB | 1 :1000 (WB) | Millipore | AB5603 (RRID:AB_2286686) |
| Anti-Sox9 (D8G8H), rabbit mAB | 1:1000 (WB); 1:500 (IF) | Cell Signaling Technology | 82630 (RRID:AB_2665492) |
| Anti-α-Tubulin, mouse mAB | 1:10000 (WB) | Sigma Aldrich | T9026 (RRID:AB_477593) |
| Anti-Tubulin b3 (TUBB3), mouse mAB | 1:500 (IF) | Biolegend | 801201 (RRID:AB_2313773) |
| Secondary antibodies | | | |
| **Name** | **Dilution** | **Source** | **Catalog number** |
| Anti-mouse IgG Alexa Fluor 647 | 1:1000 | Invitrogen | A-21245 |
| Anti-mouse IgG Alexa Fluor Plus 488 | 1:1000 | Invitrogen | A-32723 |
| Anti-mouse IgG HRP-linked | 1:5000 | Cell Signaling Technology | 7076S |
| Anti-rabbit IgG Alexa Fluor 488 | 1:1000 | Invitrogen | A-11034 |
| Anti-rabbit IgG HRP-linked | 1:5000 | Cell Signaling Technology | 7074S |
| Donkey anti-rabbit IgG (H + L), CF647 | 1:1000 | Sigma | SAB4600177 |
| Goat anti-rabbit IgG (H + L), Alexa Fluor 555 | 1:1000 | Invitrogen | A-21428 |
| Goat anti-chicken IgY (H + L) secondary antibody, Alexa Fluor 488 | 1:1000 | Invitrogen | A-11039 |
| Oligonucleotides and other sequence-based reagents | | | |
| **shRNAs** | **Sequence** | | **Targeting site** |
| Mouse_Sox10_sh1 | CCACGAGGTAATGTCCAACAT | | Coding region (TRCN0000018985) |
| Mouse_Sox10_sh2 | TTGCTCCAGCGATACCTTAAT | | 3' UTR (TRCN0000244290) |
| Non-targeting shRNA | CAACAAGATGAAGAGCACCAA | | NA |
| **sgRNAs** | **Sequence** | | |
| Human_SOX10_sg1 | ATTCAGGCTCCGTCCTAACG | | |
| Human_SOX10_sg2 | CGAGCTGGACCGCACACCTT | | |
| Human_SOX10_sg3 | AGTCTCGGGCTGTCCGGCCA | | |
| Negative control | GCGCCAAACGTGCCCTGACGG | | |
| **qPCR primers** | **Forward sequence** | | **Reverse sequence** |
| Human-SOX10 | CTTTCTTGTGCTGCATACGG | | AGCTCAGCAAGACGCTGG |
| Human-SOX9 | AGCGAACGCACATCAAGAC | | CTGTAGGCGATCTGTTGGGG |
| Human-LRIG1 | GTGTCATCACCAACCACTTTGGC | | GCAATCTGAGGGTTTGGGTGAC |
| Human-NRARP | CAAGGGCAACACGCAGGAGCT | | CCGAACTTGACCAGCAGCTTCA |
| Human-HES5 | TCCTGGAGATGGCTGTCAGCTA | | CGTGGAGCGTCAGGAACTGCA |
| Human-TBP | GAACCACGGCACTGATTTTC | | CCCCACCATGTTCTGAATCT |
| Mouse Sox10 | GTGCCAGCAAGAGCAAGCCG | | CTGCCTTCCCGTTCTTCCGCC |
| Mouse-Sox9 | CACACGTCAAGCGACCCATGAA | | TCTTCTCGCTCTCGTTCAGCAG |
| Mouse-Lrig1 | TTCAGCCAACGCTACCCTCACA | | TAAGCCAGGTGATGCGTGGTGT |
| Mouse-Nrarp | CAGACAGCACTACACCAGTCAG | | CCGAAAGCGGCGATGTGTAGC |
| Mouse-Hes5 | CCGTCAGCTACCTGAAACACAG | | GGTCAGGAACTGTACCGCCTC |
| Mouse-Tbp | ACCGTGAATCTTGGCTGTAAAC | | GCAGCAAATCGCTTGGGATTA |
| Public datasets | | | Source |
| Brain tumor stem cell bulk RNA-seq | | | Marques et al, 2021 |
| TCGA-GBM (HG-U133A) | | | TCGA |

**Drug-screening compounds**

| Cat no. | Name | Pathways | Target | Receptor |
|---|---|---|---|---|
| Drug-screening compounds | | | | |
| Cat no. | Name | Pathways | Target | Receptor |
| T1952 | MK-2206 dihydrochloride | Cytoskeletal signaling; PI3K/Akt/mTOR signaling | Akt | Akt1; Akt2; Akt3 |
| T3132 | SC66 | Cytoskeletal signaling; PI3K/Akt/mTOR signaling | Akt inhibitor | Akt |
| T2492 | Perifosine | Cytoskeletal signaling; PI3K/Akt/mTOR signaling | Akt inhibitor | Akt |
| T4489 | AZD26 | PI3K/Akt/mTOR signaling; cytoskeletal signaling | Akt inhibitor | Allosteric Akt |
| T1961 | Vistusertib | Cytoskeletal Signaling; PI3K/Akt/mTOR signaling; MAPK | Akt inhibitor; mTOR inhibitor; PI3K inhibitor; S6 Kinase inhibitor | P-Akt (S473); mTOR; PI3Kα; pS6 (S235/236) |
| T1936 | Alectinib | Angiogenesis; tyrosine kinase/adaptors; tyrosine kinase/adaptors | ALK inhibitor; tyrosine kinases inhibitor; VEGFR inhibitor | ALK; ALK (F1174L); ALK (R1275Q); INSR; VEGFR2 (KDR) |
| T1661 | Crizotinib | Angiogenesis; tyrosine kinase/adaptors | ALK; c-Met/HGFR | ALK; c-Met |
| T16156 | MT 63-78 | PI3K/Akt/mTOR signaling | AMPK; mTOR | AMPK;mTORC1 |
| T2235 | Dactolisib | DNA damage/DNA repair; PI3K/Akt/mTOR signaling | ATM/ATR; mTOR; PI3K | ATR; mTOR (p70 S6K); p110α; p110γ; p110δ |
| T2509 | Tozasertib | Cell cycle/checkpoint; chromatin/epigenetic | Aurora Kinase | Aurora A; Aurora B; Aurora C |
| T6315 | MLN8054 | Cell cycle/checkpoint; metabolism; angiogenesis; GPCR/G protein; chromatin/epigenetic; stem cells | Aurora kinase inhibitor; casein kinase inhibitor; PKA inhibitor; Src inhibitor | Aurora A; Aurora B; CK2; PKA; Lck |
| T6458 | CYC116 | Cell cycle/checkpoint; angiogenesis; PI3K/Akt/mTOR signaling; chromatin/epigenetic; tyrosine kinase/adaptors; MAPK | Aurora Kinase inhibitor; CDK inhibitor; FLT inhibitor; S6 Kinase inhibitor; VEGFR inhibitor | Aurora A; Aurora B; CDK2/CyclinE; CDK9/CyclinT; FLT3; p70 S6K; VEGFR2 |
| T2358 | ENMD2076 | Cell cycle/checkpoint; angiogenesis; apoptosis; chromatin/epigenetic; tyrosine kinase/adaptors | Aurora kinase inhibitor; c-RET inhibitor;FLT inhibitor; Src inhibitor; VEGFR inhibitor | Aurora A; RET; FLT3; Src; VEGFR3/FLT4; Src |
| T2378 | RGB-286638 free base | Cell cycle/checkpoint | CDK inhibitor | CDK1/cyclinB1; CDK2/cyclinE; CDK3/cyclinE; CDK4/cyclin D1; CDK9/cyclinT1; p35-CDK5 |
| T2029 | Bohemine | Cell cycle/checkpoint | CDK inhibitor | CDK |
| T2247 | KenPaullone | PI3K/Akt/mTOR signaling; cell cycle/checkpoint; stem cells | CDK inhibitor; GSK-3 inhibitor | CDK1/CyclinB; CDK2/CyclinA; CDK2/CyclinE; CDK5/p35; GSK-3β |
| T1778 | AT 7519 hydrochloride salt | PI3K/Akt/mTOR signaling; cell cycle/checkpoint; stem cells | CDK inhibitor; GSK-3 inhibitor | CDK2/CyclinA; CDK4/Cyclin D1; CDK5/p35; CDK9/CyclinT; GSK-3β |
| T4327 | Prexasertib dihydrochloride | Cell cycle/checkpoint; PI3K/Akt/mTOR signaling; MAPK | Chk; S6 Kinase | Chk1;Chk2;RSK |
| T4301 | AD80 | Tyrosine kinase/adaptors; MAPK; angiogenesis; PI3K/Akt/mTOR signaling; apoptosis | c-RET; Raf; Src; S6 Kinase | RET; RAF, SRC, S6K;RET V804M;RET V804L |
| T3458 | EED226 | Chromatin/epigenetic | DNA methyltransferase inhibitor; Histone methyltransferase inhibitor | EED; PRC2 |
| T2377 | XL147 analogue | DNA damage/DNA repair; PI3K/Akt/mTOR signaling | DNA-PK inhibitor; PI3K inhibitor | DNA-PK; PI3Kα; PI3Kβ; PI3Kγ; PI3Kδ |
| T1826 | Voxtalisib Analogue | DNA damage/DNA repair; PI3K/Akt/mTOR signaling | DNA-PK inhibitor; PI3K inhibitor | DNA-PK; PI3Kα; PI3Kβ; PI3Kγ; PI3Kδ |
| T14073 | A-485 | Chromatin/epigenetic | Epigenetic Reader Domain; Histone Acetyltransferase | CBP/p300;histone acetyltransferase |
| T2608 | CHIR98014 | PI3K/Akt/mTOR signaling; angiogenesis; stem cells; tyrosine kinase/adaptors; MAPK | FGFR inhibitor; GSK-3 inhibitor; S6 Kinase inhibitor; Src inhibitor | bFGFR; GSK-3α; GSK-3β; p70 S6K; c-Src |

**Drug-screening compounds**

| Cat no. | Name | Pathways | Target | Receptor |
|---|---|---|---|---|
| T2310 | CHIR99021 | PI3K/Akt/mTOR signaling; stem cells | GSK-3 | GSK-3α; GSK-3β |
| T1741 | AZD1080 | PI3K/Akt/mTOR signaling; stem cells | GSK-3 inhibitor | GSK-3α; GSK-3β |
| T1755 | LY2090314 | PI3K/Akt/mTOR signaling; stem cells | GSK-3 inhibitor | GSK-3α; GSK-3β |
| T3067 | Tideglusib | PI3K/Akt/mTOR signaling; stem cells | GSK-3 inhibitor | GSK-3β |
| T1881 | AR-A014418 | PI3K/Akt/mTOR signaling; stem cells | GSK-3 inhibitor | GSK-3β; GSK-3β |
| T1957 | AZD2858 | PI3K/Akt/mTOR signaling; stem cells | GSK-3 inhibitor | GSK-3 |
| T6187 | TDZD8 | PI3K/Akt/mTOR signaling; stem cells | GSK-3 inhibitor | GSK-3β |
| T6059 | GSK343 | Chromatin/epigenetic | Histone methyltransferase inhibitor | EZH1; EZH2 |
| T3057 | UNC1999 | Chromatin/epigenetic | Histone methyltransferase inhibitor | EZH1; EZH2 |
| T1788 | EPZ6438 | Chromatin/epigenetic | Histone methyltransferase inhibitor | EZH2; EZH2 |
| T4583 | (R)-PFI-2 hydrochloride | Chromatin/epigenetic | Histone methyltransferase inhibitor | SETD7 |
| T1841 | UNC0379 | Chromatin/epigenetic | Histone methyltransferase inhibitor | SETD8 |
| T2243 | Serdemetan | Apoptosis | Mdm2 inhibitor; p53 activator | HDM2; Mdm2; p53 |
| T6159 | LY-2584702 free base | PI3K/Akt/mTOR signaling | mTOR | mTOR (p70 S6K) |
| T1859 | AZD8055 | PI3K/Akt/mTOR signaling | mTOR | mTOR |
| T2706 | Palomid 529 | PI3K/Akt/mTOR signaling | mTOR inhibitor | mTORC1; mTORC2 |
| T1838 | INK 128 | PI3K/Akt/mTOR signaling | mTOR inhibitor; PI3K inhibitor | mTOR; mTOR; PI3Kα; PI3Kγ; PI3Kδ |
| T3153 | Serabelisib | PI3K/Akt/mTOR signaling | mTOR inhibitor; PI3K inhibitor | mTOR; p110α; p110β; p110γ; p110δ |
| T1916 | Apitolisib | PI3K/Akt/mTOR signaling | mTOR inhibitor; PI3K inhibitor | mTOR; p110α; p110β; p110γ; p110δ |
| T2357 | GSK1059615 | PI3K/Akt/mTOR signaling | mTOR inhibitor; PI3K inhibitor | mTOR; PI3Kα; PI3Kβ; PI3Kγ; PI3Kδ |
| T1861 | Omipalisib | PI3K/Akt/mTOR signaling | mTOR inhibitor; PI3K inhibitor | mTORC1; p110α; p110β; p110γ; p110δ |
| T2619 | CH5132799 | PI3K/Akt/mTOR signaling | mTOR inhibitor; PI3K inhibitor | mTOR; PI3Kα; PI3Kβ; PI3Kγ; PI3Kδ |
| T1970 | Gedatolisib | PI3K/Akt/mTOR signaling | mTOR inhibitor; PI3K inhibitor | mTOR; PI3Kα; PI3Kγ |
| T2265 | PI3K-IN-2 | Others; PI3K/Akt/mTOR signaling; MAPK | Others inhibitor;S6 Kinase inhibitor;PI3K;mTOR | pPKB; pS6;PI3Kα;PI3Kβ;mTOR;PI3Kγ; PI3Kδ |
| T1827 | Buparlisib | PI3K/Akt/mTOR signaling | PI3K | p110α; p110β; p110γ; p110δ; Vps34 |
| T1994 | Pictilisib | PI3K/Akt/mTOR signaling | PI3K | p110α; p110β; p110γ; p110δ |
| T2672 | TG100-115 | PI3K/Akt/mTOR signaling | PI3K inhibitor | PI3Kα; PI3Kβ; PI3Kγ; PI3Kδ |
| T2365 | Pilaralisib | PI3K/Akt/mTOR signaling | PI3K inhibitor | PI3Kα; PI3Kβ; PI3Kγ; PI3Kδ |
| T2073 | GSK2636771 | PI3K/Akt/mTOR signaling | PI3K inhibitor | PI3Kβ |
| T6168 | ZSTK474 | PI3K/Akt/mTOR signaling | PI3K inhibitor | PI3K; PI3Kα; PI3Kβ; PI3Kγ; PI3Kδ |
| T1988 | Duvelisib | PI3K/Akt/mTOR signaling | PI3K inhibitor | PI3Kα; PI3Kβ; PI3Kγ; PI3Kδ |
| T1999 | Taselisib | PI3K/Akt/mTOR signaling; metabolism | PI3K inhibitor; Carbonic Anhydrase | PI3Kα; PI3Kβ; PI3Kγ; PI3Kδ;C2β |
| T2008 | LY294002 | PI3K/Akt/mTOR signaling; DNA damage/DNA repair | PI3K; DNA-PK | p110α; p110β; p110δ;DNA-PK |
| T6250 | H 89 2HCl | GPCR/G protein; PI3K/Akt/mTOR signaling; MAPK | PKA; S6 kinase | PKA; S6K1 |
| T3518 | GSK269962A | Cell cycle/checkpoint; cytoskeletal signaling; PI3K/Akt/mTOR signaling; stem cells; MAPK | ROCK inhibitor; S6 kinase inhibitor | MSK1; ROCK1; ROCK2; RSK1 |
| TN1712 | Gossypin | PI3K/Akt/mTOR signaling; MAPK | S6 kinase | RSK2 |
| T6S1302 | Carnosol | PI3K/Akt/mTOR signaling; MAPK | S6 kinase | RSK2 |
| T22422 | S6K-18 | PI3K/Akt/mTOR signaling; MAPK | S6 kinase | S6K1 |
| T2002 | PF4708671 | PI3K/Akt/mTOR signaling; MAPK | S6 kinase inhibitor | p70 S6K |
| T1746 | LY-2584702 tosylate | PI3K/Akt/mTOR signaling; MAPK | S6 kinase inhibitor | p70 S6K |

| Drug-screening compounds | | | | |
|---|---|---|---|---|
| Cat no. | Name | Pathways | Target | Receptor |
| T6877 | LJH685 | PI3K/Akt/mTOR signaling; MAPK | S6 kinase inhibitor | RSK1; RSK2; RSK3 |
| T6878 | LJI308 | PI3K/Akt/mTOR signaling; MAPK | S6 kinase inhibitor | RSK1; RSK2;RSK3 |
| T4488 | GSK25 | PI3K/Akt/mTOR signaling; PI3K/Akt/mTOR signaling; MAPK | S6 kinase inhibitor; mTOR inhibitor | RSK1; mTOR (p70 S6K) |
| T5428 | BIX 02565 | PI3K/Akt/mTOR signaling; autophagy; MAPK | S6 kinase; LRRK2 | RSK2;LRRK2 |
| T2510 | Galunisertib | Stem cells | TGF-beta/Smad | TβRI |
| T1293 | Gatifloxacin | DNA damage/DNA repair | Topoisomerase inhibitor | Topo IV |
| T1878 | XAV939 | Stem cells; cytoskeletal signaling | Wnt/beta-catenin | TNKS1; TNKS2 |
| T2237 | ICG001 | Cytoskeletal signaling; stem cells; chromatin/epigenetic | Wnt/beta-catenin inhibitor; Epigenetic Reader Domain | Wnt/β-catenin; CBP |
| T2436 | GSK2801 | Chromatin/epigenetic | Epigenetic Reader Domain inhibitor | BAZ2A/2B |
| T6668 | SGC-CBP30 | Chromatin/epigenetic | CREBBP/EP300 inhibitor | CREBBP/EP300 |
| T2452 | C646 | Chromatin/epigenetic | Histone acetyltransferase inhibitor | p300/CBP |
| T6133 | Remodelin | Chromatin/epigenetic | Histone acetyltransferase inhibitor | Acetyltransferase NAT10 |

## Cell culture

Patient-derived tumorspheres NCH421k, NCH441 and NCH644 were previously characterized and provided by Christel Herold-Mende (Campos et al, 2010). Cells were maintained in stem-like conditions in DMEM/F12 (Gibco,11330-057) containing 1× B27 (without vitamin A) (Gibco, 12587101), 2 mM L-glutamine, 20 ng/ml Epidermal growth factor (Gibco, PHG0311), 20 ng/ml fibroblast growth factor 2 (PeproTech, 100-18B) and 1% Pen/Strep. Cells were maintained as spheres in suspension culture vessels. Large spheres were dissociated with Accutase at 37 °C with occasional agitation Mouse glioblastoma cell line mGB1 was provided by P Angel and was maintained in similar culture conditions with the addition of 1× N2 (Gibco, 17502048). All cell lines were kept in a 5% CO$_2$ humidified incubator at 37 °C.

## SOX10 knockdown

In mGB1 cells, Sox10 was stably knocked down using short hairpin RNA (shRNA) as previously described (Wu et al, 2020). Briefly, non-overlapping target sequences of shRNA sequences (sh1: TRCN0000018985; sh2: TRCN0000244290; Reagents and Tools Table) were cloned into pLKO.1-puro (Sigma, SHC002). For the delivery of shRNAs, lentiviruses of pLKO.1-non target control, sh1 and sh2 were produced and transduced into mGB1 cells. Transduced cells were selected with 1 μg/ml puromycin for 48 h and the KD efficiency was checked by both qPCR and western blot.

In human GSCs, SOX10 was stably knocked down using CRISPR-interference method (CRISPRi) as previously described (Wu et al, 2020). Single clones stably expressing pHR-SFFV-dCas9-BFP-KRAB (Addgene, 46911) were expanded and transduced with single guide RNA (sgRNA; Reagents and Tools Table) plasmids previously described. To knock down SOX10 expression, NCH441 and NCH644 stable clones stably expressing dCas9 were transduced with sgRNA plasmids. Transduced cells were selected with 1 μg/ml

puromycin for 48 h and the KD efficiency was checked by both qPCR and western blot.

## RNA extraction and reverse transcription quantitative PCR (RT-qPCR)

Cell pellets were first collected, and total RNA was extracted with the RNAeasy Mini Kit and reverse-transcribed into complementary DNA (cDNA) using Quantitect Reverse Transcription Kit (Qiagen, 205311), following the manufacturer's instructions. For mRNA quantification, SYBR-green-based qPCR was performed using 1× primaQuant SYBR Green reagent (Steinbrenner, SL-9902B). Primers used in qPCR are listed in the Reagents and Tools Table.

## Western blot

Cells were washed with ice-cold PBS and were lysed in RIPA lysis buffer supplemented with 1× cOmplete™, Mini EDTA-free Protease Inhibitor Cocktail and 1× phosphatase inhibitor phosSTOP. A total of 5 μg of protein samples were loaded and resolved on 4–12% of NuPAGE™ Bis-Tris Protein gels (Life Technologies, NP0335). The proteins were then transferred onto PVDF membrane. The membranes were blocked in 5% skimmed milk or 5% bovine serum albumin, before incubation with primary antibodies overnight (SOX10, SOX9, SOX2, NICD, tubulin). Corresponding horseradish peroxidase (HRP) conjugated secondary antibodies were then added. ECL substrates (Life Technologies, 32132) were then added and were subsequently detected by light-sensitive films. Alpha-tubulin served as the loading control. Primary and secondary antibodies used for Western blots were summarized in the Reagents and Tools Table.

## In vivo study

Animal experiments in this study were performed in accordance with relevant ethical regulations and were approved by the

Regierungspräsidium Karlsruhe, Germany (reference no. G-156-15) and in Shanghai, China. *Sox10* was knocked down in mGB1 cells as described above. Prior to intracranial injection, adult C57B6/J mice (8 weeks female) were ordered from Janvier and were allowed to acclimatize in the animal housing facility. For intracranial injection, 200,000 cells (Ctrl and KD cells, in 1 μl volume) were injected into adult C57B6/J mice (8 weeks female) brain under anesthesia with isoflurane. For drug studies, luciferase-expressing mGB1-KD cells were used and injected as indicated above. Tumor growth for single-cell analysis, tumor growth was monitored by MRI scanning 1 month after injection. For drug studies, tumor growth was monitored by bioluminescence imaging by i.p. injection of D-luciferin. Animals were randomized into treatment groups based on estimated tumor size prior to treatment start. Sample or treatment blinding was not applied. The sample size was determined by simulation using R analysis software. In the combination treatment group, the drugs (100 mg/kg Fimepinostat + 10 mg/kg LY411575, p.o.) were delivered on alternating days for five cycles (total of 10 days).

## Single-cell library preparation and sequencing

Mice with large tumor sizes (above 50 μl as measured by T2 MRI) in both groups were sacrificed and included in this study (4 mice from KD, 3 mice from Ctrl). After euthanization with carbon dioxide, tumor areas were carefully resected to ensure minimal inclusion of normal brain tissue. Tumors were then finely minced and dissociated using Tumor Dissociation Kit, mouse using gentleMACS dissociator (Miltenyi Biotech, 130-096-730), according to manufacturers' instructions. The cell pellets were then resuspended in 1% BSA/PBS and were sorted on BD FACSAria™ Fusion based on GFP-positivity to isolate tumor cells. They were then subject to single-cell capture and library preparation in the singe cell Open Lab in DKFZ, Heidelberg. Single-cell libraries were prepared using Chromium Next GEM Single Cell 3′ GEM, Library & Gel Bead Kit v3.1 (10X P.N.: 1000128), closely following the standard protocols recommended by 10X Genomics. Libraries were quantified using the Qubit dsDNA H.S. Assay kit. Equal amounts of the libraries were pooled and sequenced on NovaSeq 6000 with Paired-end (28 + 94 bp) S2 setup at High Throughput Sequencing Unit of the Genomics and Proteomics Core Facility at the German Cancer Research Center (DKFZ, Heidelberg). For the single-cell study in the drug treatment groups, single-cell RNA-Seq libraries were prepared using SeekOne® MM Single-cell 3' library preparation kit (SeekGene, K00104-04), sequenced on Illumina NovaSeq 6000 with PE150 read length.

## Single-cell data analysis

Raw reads in FASTQ format were aligned to the mouse reference genome mm10 version 3.0.0 by conducting the count pipeline within the 10x Cell Ranger software. For each sample, a gene-cell matrix of unique molecular identifier (UMI) counts was generated. The UMI matrices were passed to R using the *Read10x()* function from the Seurat package version 3.2.3 (Stuart et al, 2019), and were converted into a Seurat object via the *CreateSeuratObject()* function, allowing for subsequent filtering, normalization and downstream analyses. Cells with feature counts of 200 or less and cells containing 10% or more mitochondrial RNA, defined as features starting with the character string "mt-", were filtered out.

Subsequently, the count matrix was normalized using *NormalizeData()* function: Feature counts for each cell were divided by its total counts, multiplied by 10,000.

The quality of the scRNA-seq data of each sample was evaluated by examining the metadata table of the 10× pipeline, as well as analyzing the number of genes and features per cell and the percentage of mitochondrial RNA. The distribution of cells coming from the same group (e.g., *Sox10*-KD tumor cells) was further assessed in the UMAP space to estimate batch effects. To determine the differentially expressed genes (DEGs) of each cluster (Dataset EV1), FindAllMarkers () function from the Seurat package was used. The significance of the DEGs was computed using Wilcoxon tests with an adjusted (Bonferroni correction) *P* value of 0.05. Gene signature activity in single cells was evaluated using AUCell (v.1.4.1).

## TMZ treatment in glioblastoma cell lines

NCH421k and NCH441 were seeded at 100,000 cells per well on six-well plates. Cells were treated with DMSO or temozolomide (TMZ, Sigma, T2577) at 25 μM or 50 μM. TMZ was refreshed every 3–4 days before harvesting protein for WB analyses.

For TMZ pre-treatment prior to combination treatment (Fig. EV4), NCH421k and NCH441 were treated with 50 μM TMZ for 7 days. Then, the cells were dissociated with Accutase, washed and seeded at a density of 10,000 cells/well into a 96-well plate either with LY411575 (4 μM) or without. One day after seeding, Fimepinostat was added to the cells at the indicated concentrations. Forty-eight hours after Fimepinostat treatment, the viability was assessed using the CellTiter-Glo Reagent.

For inducing TMZ-recovered cells, NCH421k and NCH441 were seeded at 100,000 cells per well on six-well plates. TMZ was refreshed every 3–4 days, and viable cells were counted using the TC20 automated cell counter (Biorad #1450102) every week. Tumor cells treated with DMSO reached confluence around 10–14 days after seeding and were counted for 2 weeks, whereas cells treated with TMZ were counted for 5 weeks before temozolomide was withdrawn.

## TMZ treatment in cerebral organoid models

NCH644 cells were first transduced with pLenti-PGK-V5-GFP-LUC-Neo. A total of 100,000 NCH644-GFP-luc cells were co-cultured with iPSC-induced cerebral organoids (Linkous et al, 2019). Tumor growth was monitored by bioluminescence signals (BLI). Total proton flux was quantified using the Living Image® software. BLIs before DMSO or 100 μM of TMZ was added to the human cerebral organoid glioma were used as baseline and were measured every 3 days. To analyze the expression of SOX10 in tumor cells after TMZ treatment, tumor cells were harvested and stained with viability dye at 1:500 dilution in PBS for 30 min at 4 °C. Cells were then fixed with 4% PFA (methanol free) for 15 min at R.T. in the dark. Cells were washed with FACS buffer and incubated with permeabilization and blocking buffer (5% normal goat serum in 0.5% Triton X-100) for 1 h at room temperature. After washing with FACS buffer, cells were incubated with rabbit anti-human/mouse SOX10 antibody at 1:500 dilution and chicken anti-GFP antibody at 1:500 dilution in permeabilization and blocking buffer for 1 h at room temperature. Cells were then washed with FACS buffer and stained with conjugated secondary antibodies (anti-rabbit AF647, 1:2000 and anti-chicken AF488, 1:2000) for 30 min at room temperature. Finally, cells were washed and resuspended in FACS

buffer and proceeded to flow cytometry analysis. SOX10 expression levels were measured as the median fluorescence intensity (AF647) of the viable, GFP-positive cells. Samples stained with only secondary antibodies were used as a negative control.

## Public data analysis

Normalized BTSC and TCGA-GBM data were downloaded from their respective sources (Reagents and Tools Table). Top 25% and bottom 25% of the dataset were classified as *SOX10*-high and *SOX10*-low samples, respectively. Differentially expression analyses were performed using limma package (contrast= *SOX10*-low/*SOX10*-high). Differentially regulated genes, defined as *P*.adj <0.05, were sorted by their log2 fold change and gene set enrichment analyses (GSEA) were performed using the ClusterProfiler package (Yu et al, 2012) against curated genesets. For TCGA-GBM data, only IDH-wildtype primary tumor samples with available *SOX10* expression information were included in the analysis, resulting in 324 samples.

## Immunofluorescence staining

For immunofluorescence staining in vitro, cells were seeded on a 12-well immunofluorescence chamber or 96-well plates coated with laminin/poly-L-lysine coating buffer (5 µl laminin + 30 µl poly-L-lysine in 2 ml PBS). Upon proper attachment of the cells, cells were fixed in 4% paraformaldehyde in PBS for 15 min at room temperature. Cells were then washed twice with PBS and permeabilized with ice-cold 0.5% Triton X-100 in PBS for 20 min. They were then blocked with 5% normal goat or donkey serum in 0.5% Triton X-100 in PBS for one hour, before incubating with primary antibodies overnight. The next day, fluorochrome-conjugated secondary antibodies and DAPI or Hoechst-33342 were added. Fluorescence images were captured using an SP8 confocal microscope (for Sox9/p27) and Zeiss Cell Observer epifluorescence microscope equipped with a 10×/0.3 PlnN Ph1 DICI objective (for Gfap/Tubb3). Image analysis software ImageJ and QuPath were used to quantify the number of p27-positive and Sox9/p27 double-positive cells in each image.

For immunostaining in mouse tissue sections, slides were deparaffinized and rehydrated with sequential immersion in xylene, a series of descending concentrations of ethanol (100%, 90%, and 70% ethanol) and finally with distilled water. Next, slides were placed in a cuvette and subjected to heat-induced antigen retrieval. Briefly, slides were immersed in sodium citrate buffer (10 mM sodium citrate, 0.05% Tween-20, pH 6.0) and heated to 100 °C for 20 min using a steam cooker. Slides were allowed to cool down to RT before washing with 0.3% Triton X-100/PBS for 20 min. Afterward, the slides were permeabilized and blocked with 5% normal serum in 0.3% Triton X-100/PBS for 1 h. Primary antibodies were diluted at appropriate concentrations using 0.3% Triton X/PBS supplemented with 1% normal serum. Slides were left in a humidified chamber at 4 °C overnight. The next day, the slides were washed with PBS-T (0.1% Tween-20 in PBS) for three times for 30 min. Fluorochrome-conjugated secondary antibodies were diluted at 1:1000 in PBS-T and were incubated in the dark at RT for 1 h. After secondary antibody incubation, slides were washed with PBS three times for 30 min. Upon the last wash, slides were treated with TrueBlack (Biotium #23007) to reduce autofluorescence according to the manufacturer's instructions. After further washing steps with PBS, slides were mounted using ProLong™ Glass

Antifade Mountant with NucBlue™ Stain (Invitrogen, P36983). Fluorescence images were captured with an SP8 confocal microscope. Primary and secondary antibodies used for immunofluorescence were summarized in the Reagents and Tools Table.

## G0/G1 cell cycle analyses by flow cytometry

To analyze the distribution of G0 and G1, cells were first spun down and singularized by accutase at 37 °C. For each stain, around 0.5 million cells were used. Cells were fixed with 70% ethanol at 4 °C overnight. The next day, cells were first rehydrated with FACS buffer (2% fetal bovine serum in PBS) and then centrifuged at 9000 rpm for 10 min. Cells were washed again with FACS buffer and were then incubated with Ki67 antibody at 1:200 dilution in FACS buffer 1 h at room temperature. After primary antibody incubation, cells were washed twice with FACS buffer and stained with AF647 conjugated secondary antibody, donkey against rabbit at 1:1000 dilution in FACS buffer in the dark for 30 min at room temperature. Cells were then washed twice with FACS buffer and stained with FxCycle™ Violet Ready Flow™ Reagent (Invitrogen, R37166) in the dark for 30 min at room temperature. Samples stained with secondary antibodies only and samples stained with FxCycle™ Violet Ready Flow™ Reagent were used as secondary controls.

## Drug screening

Mouse brain tumor stem cell mGB1 cells were dissociated with Accutase (Invitrogen, 00-4555-56) and seeded into 384-well plates at a density of 2000 cells per well using 12 channel pipettes with or without 2 µM LY411575 (Selleckchem, S2714). The next day, compounds were added at 2 µM using the BRAVO automated liquid handling platform (Agilent Technologies, G5563AA) equipped with a 96-pipette head (Agilent Technologies, G5055A). The 384-well plates were incubated at 37 °C, 5% $CO_2$ for 48 h. An equal volume of CellTiter-Glo Reagent (Promega, G7570) was added to the wells. medium of 384-well. The luminescence signal was measured by multimode plate reader (Enspire, 2300). The list of compounds used in drug screening was provided in the Reagents and Tools Table.

## Graphics

A synopsis image was created with BioRender.com.

# Data availability

RNA sequencing and data have been deposited at the European Nucleotide Archive (ENA) under the accession number PRJEB77072.

The source data of this paper are collected in the following database record: biostudies:S-SCDT-10_1038-S44319-024-00258-8.

# Peer review information

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

## Acknowledgements

The authors thank DKFZ core facilities and the NCT Molecular Precision Oncology Program for excellent technical support for High-Throughput Sequencing, Light Microscopy, Flow Cytometry, Preclinical Research, and Small Animal Imaging. We thank Norman Mack for assisting with animal experiments. This work was supported by the NCT Molecular Precision Oncology Program through project number HIPO016 and by a Shanghai Municipal Education Commission (SMEC) grant to the Shanghai Frontiers Science Center for Biomacromolecules and Precision Medicine at ShanghaiTech University.

## Author contributions

**Ka-Hou Man**: Conceptualization; Validation; Investigation; Visualization; Methodology; Writing—original draft; Writing—review and editing. **Yonghe Wu**: Conceptualization; Validation; Visualization; Methodology; Writing—original draft; Writing—review and editing. **Zhenjiang Gao**: Formal analysis; Validation; Investigation; Writing—original draft; Writing—review and editing. **Anna-Sophie Spreng**: Formal analysis; Validation; Methodology; Writing—review and editing. **Johanna Keding**: Formal analysis; Investigation; Visualization; Writing—original draft; Writing—review and editing. **Jasmin Mangei**: Formal analysis; Investigation; Writing—original draft; Writing—review and editing. **Pavle Boskovic**: Formal analysis; Investigation; Writing—original draft; Writing—review and editing. **Jan-Philipp Mallm**: Formal analysis; Investigation; Writing—original draft; Writing—review and editing. **Hai-Kun Liu**: Writing—original draft; Writing—review and editing. **Charles D Imbusch**: Formal analysis; Investigation; Writing—original draft; Writing—review and editing. **Peter Lichter**: Writing—original draft; Writing—review and editing. **Bernhard Radlwimmer**: Conceptualization; Resources; Supervision; Investigation; Methodology; Writing—original draft; Project administration; Writing—review and editing.

Source data underlying figure panels in this paper may have individual authorship assigned. Where available, figure panel/source data authorship is listed in the following database record: biostudies:S-SCDT-10_1038-S44319-024-00258-8.

## Funding

## Disclosure and competing interests statement

The authors declare no competing interests.

# Expanded View Figures

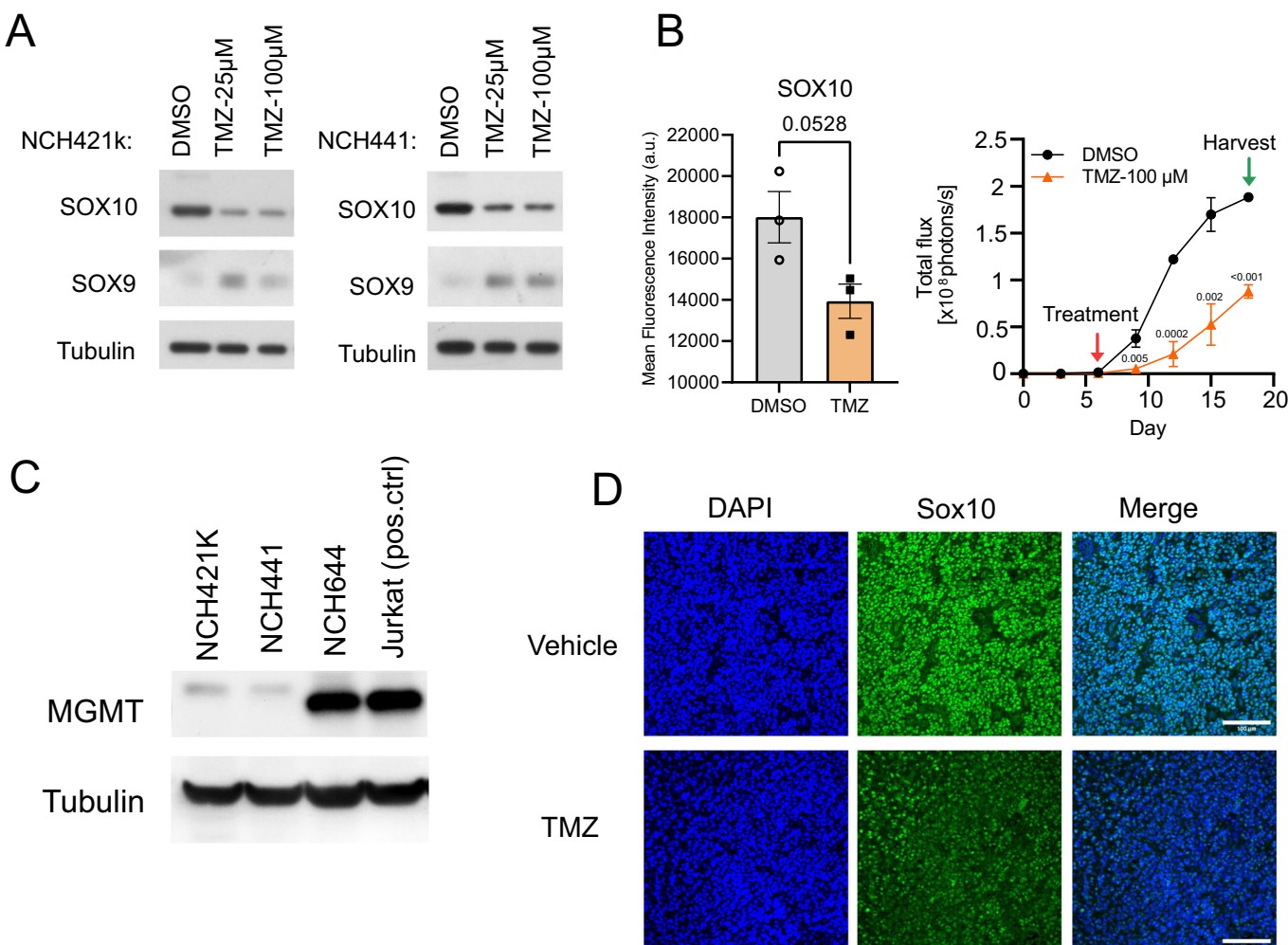

**Figure EV1. Temozolomide represses SOX10 expression.**

(A) Western blot analysis showing short-term (7-day) temozolomide treatment effects on SOX10 and SOX9 expression in NCH421k and NCH441 human glioblastoma cells. (B) Temozolomide-induced reduction of *SOX10* expression (left) and cell proliferation (right) of GFP and luciferase dual-labeled NCH644 glioblastoma cells growing in iPSC-derived cerebral organoids. Treatment with DMSO or 100 μM TMZ was started on day 6, and organoids were harvested on day 18. SOX10 expression was analyzed by recording mean fluorescence intensities by flow cytometry on day 18. Cell proliferation was monitored by bioluminescence (photon flux per second) imaging every 3 days. *n* = 3 organoids; mean ± SEM; *P* values were computed with two-tailed unpaired *T* tests. (C) Western blot analysis showing MGMT expression levels in SOX10-high glioblastoma stem cells. Protein lysate from Jurkat cells was used as the positive control for MGMT expression. (D) Immunofluorescence staining of Sox10 and Sox9 in the vehicle and TMZ-treated mice (Pdgfb/Akt RCAS glioblastoma mouse model) (Rusu et al, 2019). Scale bars = 100 μm. Related to Fig. 1.

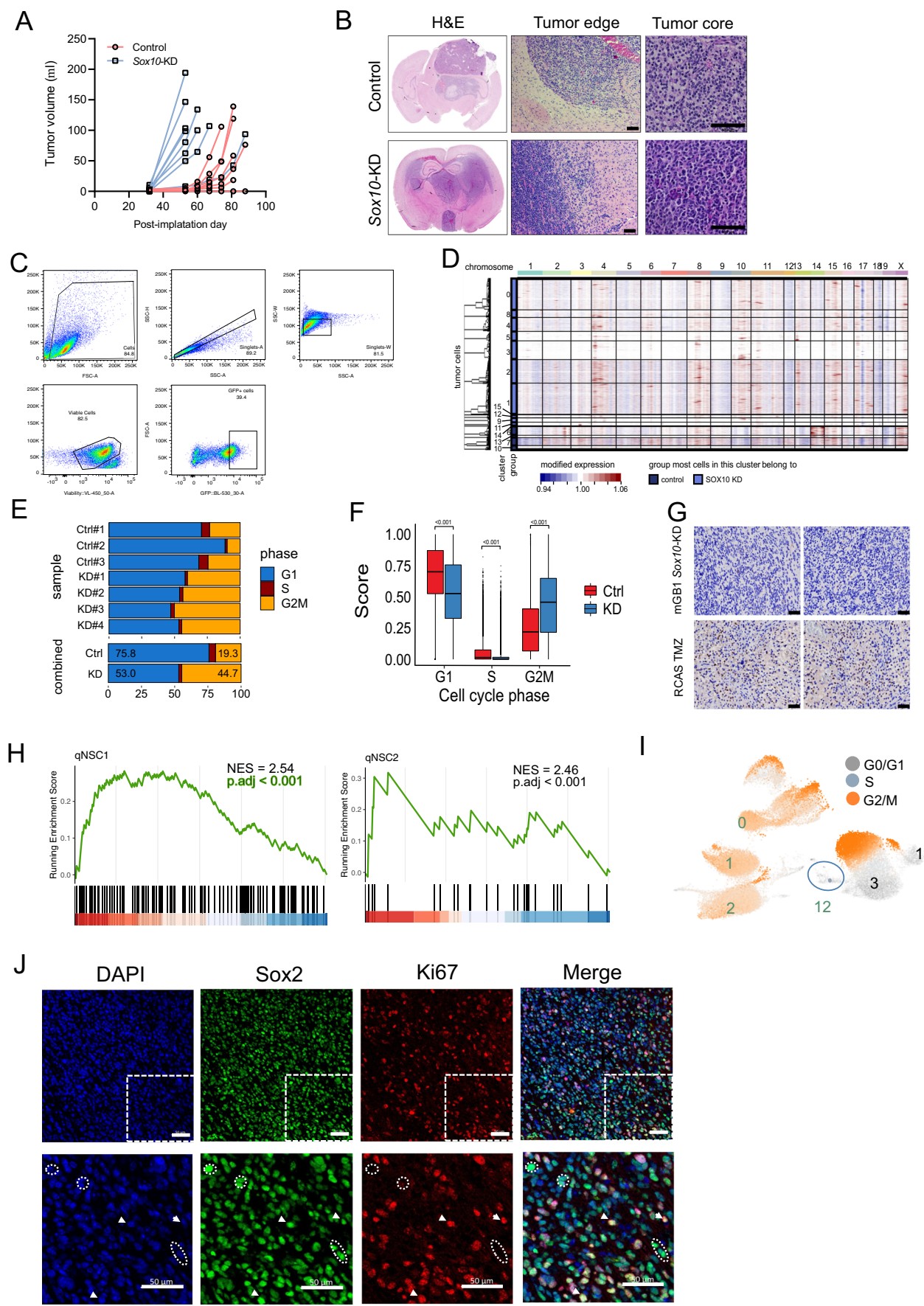

◀ **Figure EV2. Single-cell RNA sequencing analysis of control and Sox10-KD tumors.**

(A) Tumor formation in C57BL6/J mice injected with mGB1 cells expressing non-targeting shRNA (control) and shRNA against Sox10 (*Sox10*-KD) (B) H&E staining of coronal brain sections of mice bearing Ctrl and KD tumors. Note that while both tumors resemble grade 4 glioblastoma, *Sox10*-KD tumors have a distinct bi-hemispheric growth pattern, as opposed to the bulky, circumscribed growth observed in the control tumors. Scale bar = 100 μm. (C) Gating strategies for isolating GFP-positive tumor cells. (D) Analysis of the copy number variations in GFP-positive tumor cells inferred by the R-package infercnv and using GFP-negative cells as reference. Tumor cells are arranged by Seurat clusters (rows) and chromosomes (columns). Copy number gains, red; losses, blue. (E) Cell cycle phase distributions of the individual tumors. (F) Boxplots showing G1, S and G2/M phase scores in control and *Sox10*-KD tumors. For G1, S and G2M phases, $n = 6747/31090$ cells, 293/1262 cells and 1296/25182 cells from Ctrl and KD tumors, respectively. The box-and-whisker plot shows the interquartile range (box), the median cell cycle scores (line inside the box), and the range of data within 1.5 times the interquartile range from the first and third quartiles (whiskers). *P* values were computed using two-tailed Wilcoxon's rank-sum tests. ***$P < 0.001$. (G) Immunohistochemistry staining of gamma H2AX (a DNA-damage marker) in KD tumors. TMZ-treated glioblastoma mouse model (bottom) is used as a positive control for the staining. Two fields of view were shown. Scale bars = 50 μm. (H) GSEA plot showing the enrichment of qNSC gene signatures (Kalamakis et al, 2019) in cluster 12. Adjusted *P* values (*P*.adj) were calculated using the Benjamini–Hochberg method. (I) UMAP visualization of the cell cycle phases of KD tumor cells. Cluster 12 is the qNSC-like cluster. Clusters 0, 1, 2 are the fast-cycling progenitor cell-like tumor cells. Cluster 3 and 11 are the more differentiated tumor cells. (J) Immunofluorescence images of co-staining Sox2 (a tumor stem cell marker) and Ki67 (a non-G0/proliferation marker) in KD tumors. The bottom right quadrant of the images were magnified in the bottom panel to highlight the presence of slow-cycling tumor stem cells (Sox2 + /Ki67-, dotted circles) and fast-cycling tumor stem cells (Sox2 + /Ki67 + , arrows). Scale bars = 50 μm. Related to Fig. 2.

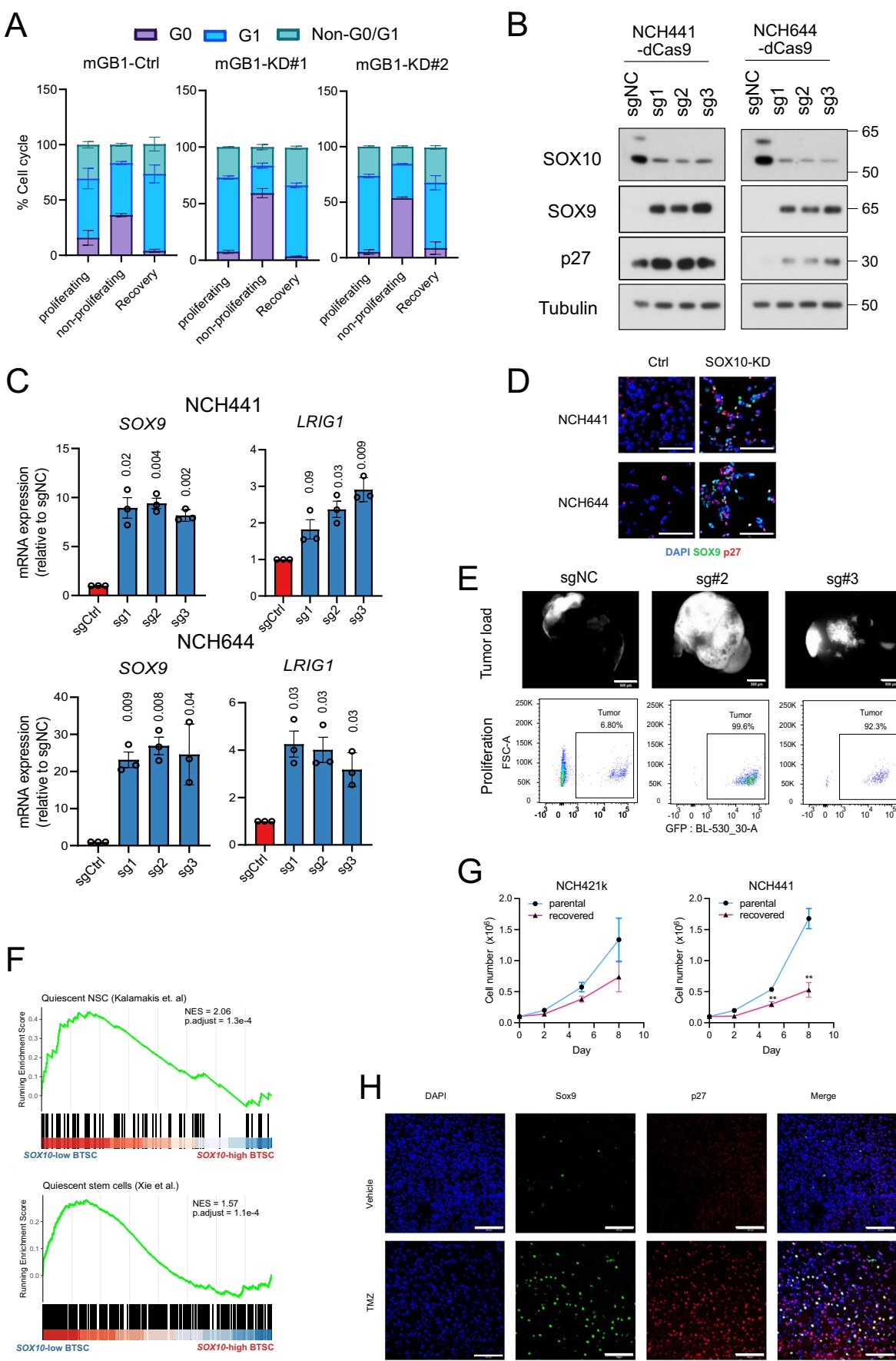

**Figure EV3.  SOX10 suppression induces quiescent stem cell features.**

(A) Bar plots showing the distribution of mGB1 control and *Sox10*-KD cells in the G0, G1 and non-G0/G1 cell cycle phases. Error bars represent mean ± SEM from three independent experiments. (B) Western blot showing the upregulation of SOX9 and p27 in control cells (sgNC) and *Sox10*-KD cells (transduced with sg1-3) in SOX10-high NCH441 and NCH644 cells. The molecular weights (in KDa) of the detected proteins are indicated on the right. (C) RNA expression of the qNSC markers *SOX9* and *LRIG1*. Mean values ± SEM relative to mGB1 control cells are shown; $N = 3$ biological replicates; P values were calculated with one-sample *T* test. (D) Immunofluorescence images of NCH441 and NCH644 control and *SOX10*-KD cells co-stained with *SOX9* and p27. Scale bars = 100 μm. (E) GFP-tagged NCH644 control (sgNC) and *SOX10*-KD (sg#2 and sg#3) co-cultured with iPSC-derived cerebral organoids. Stereoscopic fluorescence images showing the growth on day 14 after co-culture. Scale bar = 500 μm. The scatter plots below show the quantification of the tumor load by GFP flow cytometry. (F) GSEA plots showing the enrichment of slow-cycling stem cell signatures in SOX10-low vs. SOX10-high BTSCs. Adjusted *P* values (*P*.adj) were calculated using the Benjamini–Hochberg method. (G) Growth curves of NCH421k and NCH441 parental cells and cells that have recovered after TMZ treatment. Mean values ± SEM from three experiments are shown. P values were calculated using a two-tailed unpaired *t* test. (H) Sox9 and p27 immunofluorescence staining of tumor sections from mice treated with DMSO or TMZ (100 mg/kg, for 5 consecutive days). Scale bars = 100 μm. Related to Fig. 3.

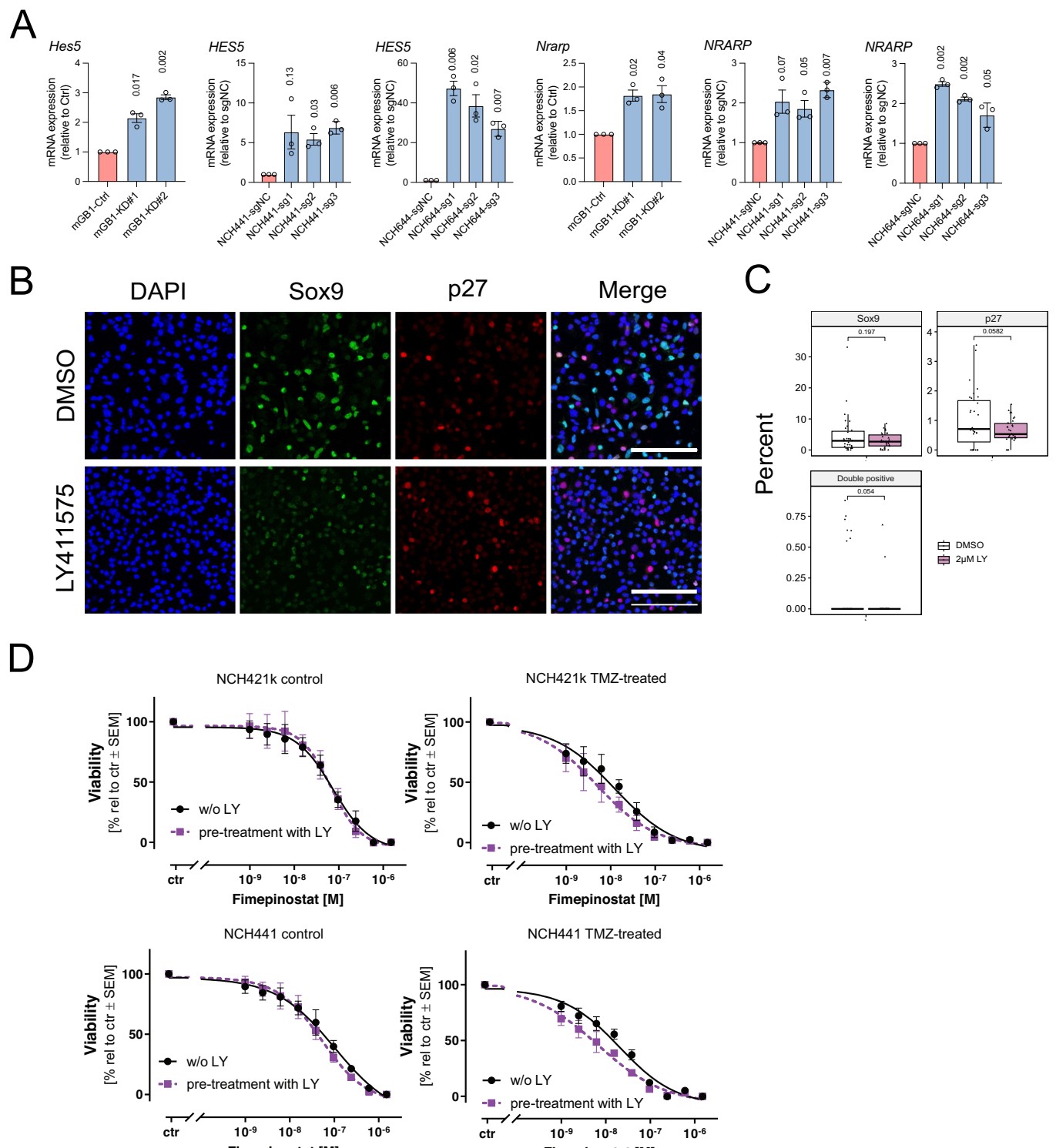

◄ **Figure EV4. Notch pathway upregulation and inhibition in Sox10-KD cells.**

(**A**) Quantification of relative RNA expression of Notch target genes (*HES5* and *NRARP*) in mouse and human glioblastoma spheroid cultures. Data are represented as mean ± SEM expression levels relative to mGB1 control cells. $n = 3$ biological replicates. *P* values were calculated using one-sample T tests. (**B**) Immunofluorescence images of mGB1 control cells stained with Sox9 and p27. Cells were cultured in reduced growth factor conditions with and without LY411575 treatment. Scale bars = 100 μm. (**C**) Boxplots showing the results of the quantifications of the percentages of Sox9-positive, p27-positive and Sox9 and p27 double-positive cells. Each dot represents the percentage of cells from an image ($n = 30$ images for each condition). The box-and-whisker plot shows the interquartile range (box), the median values (line inside the box), and the range of data within 1.5 times the interquartile range from the first and third quartiles (whiskers). *P* values were computed using two-tailed unpaired T tests. (**D**) Dose-response curves of Fimepinostat alone (black curve) and LY411575 + Fimepinostat (purple curve) in human glioblastoma cells (NCH421k and NCH441) with and without TMZ pre-treatment. Shown are the mean cell viability (%) ± SEM from three technical replicates of one experiment. Related to Fig. 4.

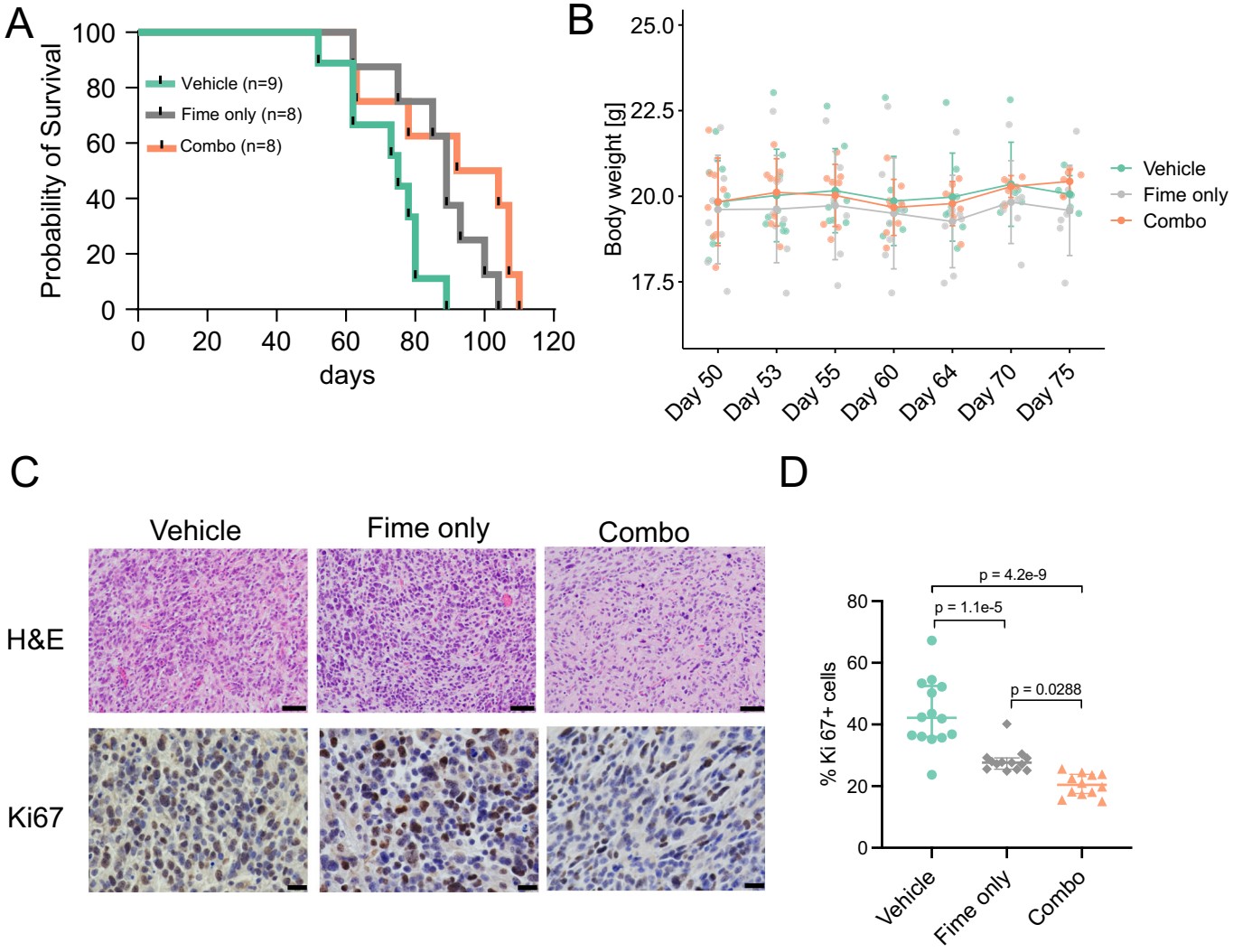

**Figure EV5. Fimepinostat treatment of mGB1 tumors in vivo.**

(A) Kaplan–Meier survival curves of mice intracranially injected with KD cells in Vehicle ($n = 9$), Fime only ($n = 8$) and Combo groups ($n = 8$), respectively. (B) Body weight of animals prior to randomization (Day 50) and during the course of the treatment (Day 53 to Day 75). On day 50, the vehicle group had $n = 9$ animals, the Fime only group had $n = 8$ animals and the Combo group had $n = 8$ animals. Shown are the mean values ± SD. (C) H&E (top) and Ki67 staining (bottom) of end-stage tumors of each treatment group. Images of the Vehicle and Combo groups were reused from Fig. 5D. Scale bars for top and bottom panels are 50 μm and 20 μm, respectively. (D) Quantification of Ki67 positive cells in (C). Shown are the median values ± interquartile ranges. *P* values were calculated with one-way ANOVA with Tukey post-hoc tests to test the level of significance among treatment groups ($n = 12$–13 images from three different animals in each group). Related to Fig. 5.

