## [Peer Review File · EMBO Reports]

SOX10 mediates glioblastoma cell-state plasticity

Ka-Hou Man, Yonghe Wu, Zhengjiang Gao, Anna-Sophie Spreng, Johanna Keding, Jasmin Mangei, Pavle Boskovic, Jan-Philipp Mallm, Hai-Kun Liu, Charles D. Imbusch, Peter Lichter, and Bernhard Radlwimmer

Corresponding author(s): Bernhard Radlwimmer (B.Radlwimmer@dkfz-heidelberg.de)

Review Timeline:

Submission Date:	4th Feb 24
Editorial Decision:	23rd Feb 24
Revision Received:	21st Jul 24
Editorial Decision:	19th Aug 24
Revision Received:	28th Aug 24
Accepted:	4th Sep 24

Editor: Esther Schnapp

Transaction Report:

Dear Dr. Radlwimmer,

Thank you for the submission of your manuscript to EMBO reports. We have now received the full set of referee reports that is pasted below.

As you will see, the referees acknowledge that the findings are potentially interesting. However, they also point out that the current set of data is not sufficiently strong to support the main conclusions, and together they raise several concerns, most of which seem to be relevant. It is clear though that not all concerns can be addressed in the timeframe of a manuscript revision. For example, referee 1 point 1 and referee 3 point 9 are not directly relevant for this study.

I would like to suggest that you draft a point-by-point response to all referee comments, to explain what you can do to address them, and what you find meaningful to do. We can then discuss the revision requirements (also in a video chat, if you like) and hopefully agree on a central set of experiments and comments that need to be addressed.

If we can agree on a set of revisions, I would like to invite you to revise your manuscript with the understanding that the referee concerns must be fully addressed and their suggestions taken on board. Acceptance of the manuscript will depend on a positive outcome of a second round of review. It is EMBO reports policy to allow a single round of major revision only and acceptance or rejection of the manuscript will therefore depend on the completeness of your responses included in the next, final version of the manuscript.

We realize that it is difficult to revise to a specific deadline. In the interest of protecting the conceptual advance provided by the work, we recommend a revision within 3 months (25th May 2024). Please discuss the revision progress ahead of this time with the editor if you require more time to complete the revisions.

- 1) A data availability section providing access to data deposited in public databases is missing. If you have not deposited any data, please add a sentence to the data availability section that explains that.
- 2) Your manuscript contains statistics and error bars based on $n=2$. Please use scatter blots in these cases. No statistics should be calculated if $n=2$.

5) a complete author checklist, which you can download from our author guidelines <https://www.embopress.org/page/journal/14693178/authorguide>. Please insert information in the checklist that is also reflected in the manuscript. The completed author checklist will also be part of the RPF.

6) Please note that all corresponding authors are required to supply an ORCID ID for their name upon submission of a revised

manuscript (<<https://orcid.org/>>). Please find instructions on how to link your ORCID ID to your account in our manuscript tracking system in our Author guidelines <<https://www.embopress.org/page/journal/14693178/authorguide#authorshipguidelines>>

I look forward to seeing a revised form of your manuscript when it is ready.

Referee #1:

Ka-Hou Man et al demonstrated the essential role of SOX10 in glioblastoma cell-state plasticity and progression. Authors proposed that SOX10 suppression mediates glioblastoma progression through NSC/developmental cell state transition and quiescent-NSC state. This work provides a new molecular mechanism for the design of tumor therapies based on phenotypic plasticity. Overall, this work is of high quality and well-organized. However, additional evidences are needed to strengthen the authors' presentations and conclusions.

Major points:

1. How does TMZ suppress SOX10 expression and upregulate SOX9 expression?
2. Does the recurrent tumors after TMZ treatment in patients preferentially exhibit NSC-like features when compared to primary tumors, phenocopying TMZ-inhibited SOX10 function on tumor state transition?
3. Sox10-KD tumors had a higher proportion of cells in the G2/M cell cycle phase. On the other hand, Sox10-KD tumors enriched a qNSC-like cell state. However, stem and progenitor cells and their progeny are in G0-like states, which cause cells to enter long- or short-term states of quiescence. Authors should analyze neural development stage combined with cell state.
4. Authors chose Fimepinostat because of its wide-ranging inhibitory effect and anti-tumor activity (Fig. 5E). The wide-ranging inhibitory effect may cause wide range damage on normal tissues. Appropriate screening of Fig. 5E inhibitors based on appropriate indicators such as less normal tissue damage or more effective tumor killing in vivo would be helpful.
5. Although Notch pathway activity was reported to be crucial in maintaining the slow-cycling stem cell state, authors did not present direct evidence that Notch signaling is downstream of SOX10-KD-mediated qNSC-like status. Rescue experiments are encouraging.
6. Fig.6B showed combination of Fimepinostat and LY411575 (Combo) inhibited tumor growth. On what extent does the single administration of Fimepinostat or LY411575 inhibit tumor growth in vivo when compared to Combo? Does Combo has a synergistic effect when compared to single treatment?
7. TMZ is the first-line chemotherapy drug for GBMs. Authors showed that TMZ treatment suppressed SOX10 expression and enriched a qNSC-like cell state. Why didn't authors explore whether LY411575 or Fimepinostat single/Combo sensitize glioblastoma cells to TMZ?

Minor points:

8. Please adjust the thickness of the scale bar to the same size in IF images.
9. Figure 1F, please show the same scaled images.
10. Figure EV1D, need a better quality image.
11. Figure EV1F, x-y axis is missing.
12. Figure 2C is an IF image, not "immunohistochemical staining" (in the main text).

Referee #2:

General Comment

The manuscript by Ka-Hou Man and colleagues investigates the effect of transcription factor SOX10 downregulation on the glioblastoma (GBM) transcriptional landscape with single-cell resolution, and treatment strategies for GBM that downregulate SOX10. SOX10 was previously identified by the same group as a determinant of oligodendrocyte cell fate specification during neural stem cell (NSC) differentiation, as a key regulator of RTK-1 (proneural) GBM subtype, and as a factor that prevents tumor progression. In this study, an association between SOX10 downregulation and the NSC state is extensively described by means of single-cell analysis of experimental tumors obtained by syngeneic transplantation of SOX10 KD murine cell lines. In SOX10 KD tumors, a quiescent NSC-like subpopulation is identified, which is further characterized by in vitro neurosphere models and found amenable to a therapeutic strategy combining Notch inhibition (capable of reinstating proliferation of quiescent NSCs) and the HDAC/PI3K inhibitor Fimepinostat. The authors attempt to establish a connection between the standard therapeutic treatment of GBM, temozolomide (TMZ), and SOX10 downregulation, to infer that the treatment effective in SOX10 KD may be useful to overcome TMZ resistance. Unfortunately, this part is largely flawed and unconvincing.

The manuscript contains abundant novel and potentially significant information, which is however often rather superficially explored. The power of single-cell analysis remains partly unexpressed due to poor data validation and interpretation; in particular, the analysis of treated tumors appears quite preliminary. The models used are insufficiently characterized and, as noticed in detail below, based on the evidence provided in this manuscript, they seem poorly representative of the TMZ response in GBM patients. This is a serious weakness that prevents from providing results with potential preclinical value. In

addition, the methods section is largely insufficient either to reproduce or to understand the data and the manuscript title, although attractive, seems unfocused on the data presented. All these flaws could however be amended. This study could be relevant for an audience interested in biological aspects of glioblastoma and, if adequately revised, also to those interested in preclinical neuroncology.

MAJOR POINTS

N. 1

Figure 1A. The methodology used for analysis of the GBM cohort is not reported in the Methods section and needs to be added. Moreover, analysis of 'MES2' is missing from the 'Nefel signature' and should be added, even in case of not significant enrichment score.

N.2

Figure 1F-G. These results are insufficient to support the conclusion that SOX10KD prevents differentiation. Expression of differentiation markers (e.g. oligodendrocytic: GalC, astrocytic: GFAP, neuronal: beta3-tub) needs to be shown. In addition, culturing cells in 10% FBS is probably not the best condition to induce differentiation, as proliferative cues may prevail. The use of lower FBS concentrations is recommended (1-2%). Along with SOX2, SOX9 and SOX10 should be shown in the western blot, as a further validation of the analysis shown in Fig. 1E

N.3.

To introduce the point that GBM therapy could impinge on SOX10 expression, the cohort analyses shown in Fig. 2D-E could be presented first in this section. As SOX9 is often mutually exclusive with SOX10, it would be interesting to see the result of SOX9 analysis as well, both in the overall cohorts and in representative IHC. However, it must be stressed that the expression profile of recurrent tumors is strongly influenced not only by TMZ therapy, but also by radiotherapy, which patients almost invariably receive. Indeed, in the literature, SOX10 downregulation has been amply correlated with the effect of ionizing radiations. If selection of cases that received TMZ only is impossible, or the related statistics cannot reach significance for the low number, evidence in favor of a meaningful relationship between SOX10 expression and prior TMZ therapy remains weak, so this analysis would be better removed.

N. 4

For in vitro and in vivo experiments, the authors use multiple models including original mouse and human GBM neurospheres and organoids that are defined as 'previously characterized'. However, the main characteristics of these models should be reported again, including the essential genetic and molecular information, as they are crucial to interpreting the results, in particular those related to TMZ effect. A notable omission in the 'MGMT status' of the cells, i.e., the expression of the dealkylating enzyme MGMT, which repairs DNA damage inflicted by TMZ and is currently known as the main factor of resistance to TMZ therapy. It is recommended to assess MGMT protein expression and not simply MGMT promoter methylation as is routinely tested in patients. As the TMZ dose used in experiments such as those represented in Fig. 2B and EV3 is quite high (100 μ M) and able to effectively kill MGMT-negative GBM cells, the partial response observed in these models suggests that they may be MGMT-positive and thus inherently resistant to the drug. As a consequence, the effects observed after TMZ treatment with high doses can be unspecific, and anyway not representative of the TMZ doses used in patients. In summary: the MGMT expression of the models need to be shown and TMZ treatment need to be performed with dose curves (5-100 micromolar), and the results reevaluated in light of new evidence. In addition, the treatment protocols need to be clearly described in the Methods section (this important information is currently missing).

N.5

Fig. 2A-C. In TMZ-treated cells, besides SOX10 expression, it would be interesting to show SOX9 as well.

N.6

Fig. 3D. In single cell analysis, the significantly higher proportion of G2/M cells observed in SOX10-KD tumors vs. controls is interesting and should be more thoroughly interpreted (page 5). Accumulation of cells in G2/M can be consistent with faster growth but also with a late cell cycle block due to detection of irreparable DNA damage (mitotic catastrophe). Or shall we postulate a role for overexpressed p27? In addition, the percentages referring to cells in G1 and S phase should be reported.

N.7

Fig. 3E. Surprisingly, the 'Nefel signature', originally developed in for GBM single cell analysis, and applied by these authors to analyses of GBM tissues was not applied to single cells derived to SOX10-KD. Why not? Can this analysis be added?

N.8

Fig. 3F. Identification of NSC marker upregulation at RNA level needs to be accompanied by validation of protein expression in GBM tissues (e.g. by immunofluorescence). This could be conveniently shown in main Figure by relocating the analysis presented in Fig. 3G in EV. It is suggested to extract from Fig. 3G information related to cluster 12 and to integrate it with validation of protein expression in tumor tissues. As cluster 12 represents a tiny cell fraction, a spatial RNA analysis (RNAscope) could be conveniently added. Addition of these data (to be shown in main Figure) is required to robustly support the conclusion that SOX10-KD tumors include a fast-cycling and a slow-cycling NSC population.

N.9

Data presented in EV2A can not be interpreted by this reviewer, as the related methodology is not reported in the Methods section. In the figure legend the content of Q1-Q4 scatter plot areas is not indicated.

N.10

As the Results paragraph starting page 6 emphasizes that 'TMZ treatment and SOX10-KD enrich a qNSC-like state in vitro', data concerning TMZ-treated cells should be presented in the main figure. However, as in the case of Fig. 2B, it is impossible to evaluate whether the model can be representative of human GBMs that are sensitive to TMZ therapy, as information about MGMT expression is missing. After the high TMZ dose used (100 μ M, again the therapeutic schedule is missing), one would expect that MGMT negative GBM cells would be completely eradicated. The presence of residual cells suggests that these cells were likely MGMT-positive at the origin, and the toxic effect of TMZ is largely unspecific. If this is the case, the model would not be representative of human treatment, where MGMT-positive cells are fully resistant to the TMZ doses (less inferior to 100 μ M) that can reach the tumor. In summary, to show that, after a TMZ treatment mimicking human treatment, a resistant qNSC population remains that could be treated by the strategy developed in SOX10-KD tumors, it would be required to utilize two independent MGMT-negative models, treated with doses of TMZ not exceeding 50 μ M (ideally 5 μ M x 3 administrations) and analyze the cell population emerging after this selection.

N.11

If the authors intend to robustly support that TMZ-treated cells mirror SOX10-KD models, they should repeat at least the in vitro experiments of LY411575-Fimepinostat therapy in combination with TMZ treatments (in MGMT- models). The authors should investigate whether the new drug combination could be effective in cells emerging after selection by TMZ treatment.

N.12

The single-cell analysis on experimental tumors treated with LY411575-Fimepinostat appears as a technological display failing to fulfill the promise of increasing insight into the nature of glioblastoma or its evolution under therapeutic pressure. Without showing that LY411575-Fimepinostat is a candidate therapy for glioblastoma, to investigate the profile of single cells treated by this therapy is of dubious significance. The message that the relapsing tumor tend to display a mesenchymal profile, similar to tumor relapsing after conventional therapy should be further elaborated or, perhaps, better served as a starting point in future work.

MINOR POINTS

N.1

The nomenclature for the p53 gene should be updated from 'Trp53' to 'Tp53'.

N.2

Fig. 1D. The use of EGFR and ERBB3 as markers of SOX9 and SOX10 positive cells (NSC and OLC, respectively) should be more extensively explained. In Figure 1 legend the acronym NLC should be corrected in 'NSC'.

N.3

Page 7. Fig. EV4C seems wrongly cited.

Referee #3:

In this manuscript, Man et al. investigated the role of SOX10 in the GBM cellular state transition from the oligodendrocyte lineage to the neural stem cell (NSC) stage. They observed that SOX10 knockdown led to an increase in SOX9 expression, resulting in the emergence of a quiescent NSC-like cell population, a phenotype reminiscent of that induced by TMZ treatment. The authors proposed targeting the Notch signaling pathway, crucial for quiescent NSC cells, as a therapeutic strategy for SOX10-knockdown tumors. They conducted a drug screening and identified that cells treated with a Notch inhibitor showed sensitivity to the HDAC/PI3K dual inhibitor Fimepinostat. This study presents interesting and findings with novelty and potential clinical implications. The manuscript is generally well-written, and the data quality is generally acceptable. However, there are major concerns on the experiment design of this study. Their data is not solid to support some of the main conclusions. The manuscript's enthusiasm level is somewhat moderate to low.

1. In Figure 1E, it would be helpful to clarify whether the GFP labels tumor cells. If so, it appears that the SOX9 staining is observed in GFP-negative stromal cells, which does not support their conclusion that SOX10 KD induce SOX9 expression in tumor cells.
2. In Figure 1G, the authors should include the immunoblotting data for SOX10 and SOX9.
3. Figure 2C should address whether Sox9 expression is upregulated upon TMZ treatment.
4. For Figure 2E, the authors could strengthen the analysis to perform SOX10 staining in multiple samples and provide quantification. A scale bar should be included.

5. In Figures 2D and 2E, the authors should also compare the expression levels of SOX9 between primary and recurrent tumors.
6. In Figure 3G, why cluster 10 was removed? Additionally, clusters 4, 3, and 11, labeled as differentiated, should be further categorized into specific neural lineage differentiations, such as astrocytes, neurons, or oligodendrocytes.
7. In Figure 5F and Figure 6, to determine the synergistic effects of LY and Fimepinostat, it is essential for the authors to incorporate four distinct treatment groups: control, individual drug treatments (LY or Fimepinostat), and combination therapy (LY and Fimepinostat) for both in vitro and in vivo experiments. Moreover, in Figure 5F, they should perform the same experiment in mGB1-shRNA-control cells.
8. In Figure 6I, combo treatment induced MES-like signature. It is known that the MES-like signature is associated with immune response. Did the author observe alterations in the immune microenvironment between control and combo groups?
9. Finally, elucidating the mechanism underlying SOX10's regulation of GBM stemness is imperative to strengthen the overall conclusion of this manuscript.

To all reviewers,

Thank you very much for your thoughtful assessment of our manuscript, which we greatly appreciate. We will address your comments in detail in the point-by-point response below. To put our responses in context, we first would like to better explain our view of the essence of our data and our interpretation.

Our manuscript reports that SOX10 suppression reprograms glioblastoma towards an aggressive neural stem cell-like phenotype, including a slow-cycling, quiescent stem cell state. Based on single-cell sequencing analysis and in-vitro drug pre-screening, we designed a NOTCH1-pathway and Fimepinostat inhibitor combination treatment targeting slow-cycling stem cells and proliferating cells, which extended survival in a syngeneic glioblastoma mouse model. Single-cell sequencing analysis of treated and untreated tumors demonstrated that, although the treatment eliminated the targeted cells, it also resulted in new tumor cell populations with both quiescent and MES/injury stem cell characteristics. These cells showed reduced Notch1 expression, presumably allowing them to evade anti-Notch1 treatment. These data establish a novel mechanism of SOX10 affecting glioblastoma aggressiveness and overall survival and provide a rationale for designing tumor therapies based on single-cell analysis.

Before getting to the point-by-point responses, we would like to address three central topics.

First (**Topic 1**), we agree with Reviewer 2 that "the expression profile of recurrent tumors is strongly influenced not only by TMZ therapy but also by radiotherapy, which patients almost invariably receive." Since we do not have sufficient informative data on matched primary and recurrent glioblastoma, we cannot dissect the effects of radiotherapy and TMZ in patients. Therefore, we will follow Reviewer 2's suggestion and remove the analysis of SOX10 expression in patient tumors from the manuscript. However, we would like to point out that loss of SOX10 expression has been reported to occur in response to different cancer treatments. In glioblastoma mouse models, SOX10 downregulation has been observed after irradiation (Lau *et al*, 2015) and TMZ treatment (this study), and a TGF- β -mediated stress-signaling cascade activated by irradiation, hypoxia, and TMZ in human glioma cells (Tabatabai *et al*, 2006). In addition, SOX10 downregulation and tumor progression have been associated with BRAF inhibition in melanoma (Capparelli *et al*, 2022; Sun *et al*, 2014). Together, these studies suggest that the downregulation of SOX10 is not specific to TMZ therapy but occurs in response to different types of treatment. Our study uses TMZ to induce SOX10 downregulation and the enrichment of a slow-cycling SOX9/p27-pos cell state. We consider this an informative model in addition to SOX10-KD, but we concede that direct comparison to the human patient situation is premature.

To show that SOX10 expression is affected not just by TGF- β (and potentially other stimuli in the context of patient treatment) in addition to TMZ, we confirmed the TGF- β -induced suppression of SOX10 expression in melanoma and glioblastoma cells, including the mGB1 cells used in this study (Figure 1, below). We believe, that these published and newly generated experimental data put the relevance of glioblastoma cell MGMT status in perspective.

Figure 1 Western blotting analysis showing the effects of TGF β 1 treatment (2 ng/ml to 10 ng/ml for 5 days) on SOX10 expression in LN229 and ZH487 human glioblastoma, mGB1 mouse glioblastoma, and A375 human melanoma cells.

Second (**Topic 2**), reviewers suggested exploring whether treatment with LY411575 and Fimepinostat sensitizes glioblastoma cells to TMZ. These are relevant questions that previously have been addressed. Yahyanejad et al. (Yahyanejad *et al*, 2016) showed that treatment with the Notch1 pathway inhibitor RO4929097 sensitizes glioblastoma cells to radiation and TMZ treatment in an orthotopic mouse model. Pal et al. (Pal *et al*, 2018) demonstrated that Fimepinostat modulates DNA damage response by inhibiting radiation-induced DNA repair pathways and sensitizes high-grade glioma cells to radiation treatment in vitro and in vivo. Thus, these studies showed treatment-sensitizing effects for Fimepinostat and a gamma-secretase inhibitor such as LY411575 used by us. We are now referencing these studies in the revised manuscript. (**page 9**) We agree that testing for possible synergistic effects of the applied inhibitors with TMZ could still reveal additional relevant information; however, we think this topic distracts from the focus of our manuscript on elucidating the molecular basis of the increased tumor aggressiveness resulting from SOX10 suppression and using the gained knowledge to explore the tumors' treatment response on the single-cell level.

Third (**Topic 3**), reviewers suggested we compare all single and combination treatments in vitro and in vivo. This contrasts with our approach, in which we selected drugs to target particular tumor cell populations and validated the rationale of our approach in vitro before testing the effects of the combination treatment in vivo. We aimed our efforts at studying glioblastoma phenotypic plasticity, which we emphasize in the abstract and the introduction. From our study, we hope to gain information on the molecular mechanisms of SOX10-KD-induced progression and the response of the KD tumors to targeting specific driver-cell populations. Understanding this phenotypic plasticity will be essential for designing more effective therapies.

Therefore, we showed in vitro that Notch1 inhibition by LY411575 increased cell proliferation in the Sox10-KD but not the control group (**Fig. 4E**) and sensitized cells to Fimepinostat

antiproliferative treatment (**Fig. 4G**). In the animal experiment, we analyzed how the combination treatment affected tumor cell-state composition, confirming the treatment depleted the targeted populations (**Fig. 5G, H**) and studied its impact on survival. In addition, we identified treatment-induced cell populations that might be responsible for limiting the achieved gain in survival, providing a starting point for future explorations (**Fig. 5I**).

While we used LY411575 single treatment at sub-lethal concentrations to validate our hypothesis *in vitro*, we did not include an LY411575 single-treatment group in the animal treatment study but instead focused on validating the efficient depletion of the Notch1-positive quiescent stem cell cluster by scRNA-Seq and the survival comparison of the combination-treated vs. untreated groups. We did, however, analyze a Fimepinostat single-treatment group (**Fig. EV5** of the revised manuscript) that we had not included in the original submission because we wanted to focus on the effects of the combination treatment. The overall survival of animals in the Fimepinostat group was intermediate between the control and combination-treatment group animals. These data suggest that NOTCH1-pathway inhibition depletes NOTCH1-high cancer stem cells (as demonstrated by scRNA-Seq) and significantly prolongs survival, presumably by sensitizing the tumor cells to Fimepinostat.

Capparelli C, Purwin TJ, Glasheen M, Caksa S, Tiago M, Wilski N, Pomante D, Rosenbaum S, Nguyen MQ, Cai W *et al* (2022) Targeting SOX10-deficient cells to reduce the dormant-invasive phenotype state in melanoma. *Nat Commun* 13: 1381

Lau J, Ilkhanizadeh S, Wang S, Miroshnikova YA, Salvatierra NA, Wong RA, Schmidt C, Weaver VM, Weiss WA, Persson AI (2015) STAT3 Blockade Inhibits Radiation-Induced Malignant Progression in Glioma. *Cancer Res* 75: 4302-4311

Pal S, Kozono D, Yang X, Fendler W, Fitts W, Ni J, Alberta JA, Zhao J, Liu KX, Bian J *et al* (2018) Dual HDAC and PI3K Inhibition Abrogates NFkappaB- and FOXM1-Mediated DNA Damage Response to Radiosensitize Pediatric High-Grade Gliomas. *Cancer Res* 78: 4007-4021

Sun C, Wang L, Huang S, Heynen GJ, Prahallad A, Robert C, Haanen J, Blank C, Wesseling J, Willems SM *et al* (2014) Reversible and adaptive resistance to BRAF(V600E) inhibition in melanoma. *Nature* 508: 118-122

Tabatabai G, Frank B, Mohle R, Weller M, Wick W (2006) Irradiation and hypoxia promote homing of haematopoietic progenitor cells towards gliomas by TGF-beta-dependent HIF-1alpha-mediated induction of CXCL12. *Brain* 129: 2426-2435

Yahyanejad S, King H, Iglesias VS, Granton PV, Barbeau LM, van Hoof SJ, Groot AJ, Habets R, Prickaerts J, Chalmers AJ *et al* (2016) NOTCH blockade combined with radiation therapy and temozolomide prolongs survival of orthotopic glioblastoma. *Oncotarget* 7: 41251-41264

Reviewer 1

1. How does TMZ suppress SOX10 expression and upregulate SOX9 expression?

We unfortunately cannot answer this interesting question at the time. However, as discussed under **Topic 1** in the **Response to All Reviewers**, SOX10 downregulation might be part of a TGF-mediated stress response induced by radiotherapy, TMZ, and hypoxia in glioma and BRAF1 inhibition in melanoma. In we experimentally confirmed that SOX10 is downregulated by TGF- β treatment in glioblastoma cell lines used in this manuscript, and a melanoma cell line (Figure 1 in the **Response to All Reviewers**). TGF- β was reported to also phosphorylate and stabilize SOX9 in chondrocytes (Coricor & Serra, 2016), providing a potential explanation how TMZ could suppress SOX10 and upregulate SOX9 expression.

2. Does the recurrent tumors after TMZ treatment in patients preferentially exhibit NSC-like features when compared to primary tumors, phenocopying TMZ-inhibited SOX10 function on tumor state transition?

As discussed under **Topic 2** of our **Response to All Reviewers**, we plan to exclude analyses of patient-tumor data from the revised manuscript. Nevertheless, even though qNSC-like cells represent only a tumor cell subpopulation, we can confirm that the NSC-like "Type-1" signature is enriched in the recurrent tumors of the GLASS cohort.

Figure 2 Single sample geneset enrichment (ssGSEA) scores of Type 1 or NSC-like GB¹ of primary and recurrent tumors in the 2022 updates of the GLASS cohort of matched primary-recurrent GB pairs² (n = 114 pairs). P-value was calculated by two-tailed Wilcoxon's rank sum test.

3. Sox10-KD tumors had a higher proportion of cells in the G2/M cell cycle phase. On the other hand, Sox10-KD tumors enriched a qNSC-like cell state. However, stem and progenitor cells and their progeny are in G0-like states, which cause cells to enter long- or short-term states of quiescence. Authors should analyze neural development stage combined with cell state.

We agree with the reviewer that quiescent stem cells are expected to be in G0, whereas other populations of cancer stem- and progenitor cells can show various degrees of proliferation, e.g., transit-amplifying states. However, neural developmental stages in tumors are similar but not identical to those of normal neurodevelopment, making their detailed and informative assignment very difficult. Therefore, we analyzed cell-cycle-states in the scRNA-Seq data. The analysis revealed that the quiescent stem cell cluster 12 and the "differentiated" clusters 3 and 11 display G0 signatures, while the "progenitor" clusters 0,1, and 2 were characterized by neural/glial marker gene expression and moderately enriched G2/M signatures. We now are providing this analysis to the readers in **Fig. EV21** of the

revised manuscript. We additionally confirmed the non-proliferating stem cell status of a small population of cells by Ki67/Sox2 double staining of Sox10-KD tumor sections (**Fig. EV2J**).

4. Authors chose Fimepinostat because of its wide-ranging inhibitory effect and anti-tumor activity (Fig. 4E). The wide-ranging inhibitory effect may cause wide range damage on normal tissues. Appropriate screening of Fig. 5E inhibitors based on appropriate indicators such as less normal tissue damage or more effective tumor killing *in vivo* would be helpful.

Fimepinostat was selected based on its high sensitivity index (**Fig. 4e**; prev.5E) and broad activity. It has a favorable safety profile (Younes *et al*, 2016; now cited on **page 7**), and we did not observe significant effects on animal weight in our treatment study (**Fig. EV5B**). Thus, we do not have any indications of toxicity-related animal death. Our primary aim is to characterize SOX10-dependent phenotypic and cell-state plasticity in a syngeneic glioblastoma tumor model on the single-cell level and to explore the response of the tumors to targeting specific cell populations. In this context, we feel an additional toxicity screen of the considered inhibitors is not essential for this manuscript.

5. Although Notch pathway activity was reported to be crucial in maintaining the slow-cycling stem cell state, authors did not present direct evidence that Notch signaling is downstream of SOX10-KD-mediated qNSC-like status. Rescue experiments are encouraging.

Thank you for this insightful comment. We agree that, while showing that Sox10 KD results in the formation of a Notch+ qNSC cluster, we did not identify the upstream, Notch-activating components and mechanisms in this cluster. However, addressing this question would require not only rescue experiments but also additional, extensive screening and mechanistic analysis. To nevertheless address this important question, we are now including a discussion (**page 9**) of the findings of Wang *et al*. (Wang *et al*, 2019), who showed that NOTCH1 expression in glioma stem cells is mediated by a SOX9-SOX2 signaling axis that leads to the promoter demethylation and upregulation of NOTCH1. Given that we show that both SOX9 and SOX2 levels increase upon SOX10 downregulation, we suggest the proposed SOX9-SOX2 mechanisms could be responsible for SOX10-KD-induced NOTCH1 activation in some tumor cell populations.

6. Fig.6B showed combination of Fimepinostat and LY411575 (Combo) inhibited tumor growth. On what extent does the single administration of Fimepinostat or LY411575 inhibit tumor growth *in vivo* when compared to Combo? Does Combo has a synergistic effect when compared to single treatment?

As discussed under topic 3 in the **Response to All Reviewers**, we recorded, in parallel with the vehicle and combination treatment groups, survival data of a group of animals treated with Fimepinostat only. The survival of the Fimepinostat-treated animals was intermediate between those of the DMSO (vehicle) and LY411575+Fimepinostat (combo) treated animals. Based on these data, we cannot differentiate between additive and synergistic drug effects. We are now showing these data in **Fig. EV5**.

7. TMZ is the first-line chemotherapy drug for GBMs. Authors showed that TMZ treatment suppressed SOX10 expression and enriched a qNSC-like cell state. Why didn't authors explore whether LY411575 or Fimepinostat single/Combo sensitize glioblastoma cells to TMZ?

Finding drugs that sensitize tumor cells to TMZ was not the goal of our study. Rather, we aimed our efforts at studying glioblastoma phenotypic plasticity, to identify potential driver-cell

populations and mapping out the response of the KD tumors to the targeting of these populations. Nevertheless, we agree that adding these data would improve the manuscript. Therefore, we performed LY+Fimepinostat treatment of TMZ-pretreated and control human glioblastoma cells. The analysis, included in **Fig. EV4D** of the revised manuscript, indicates that LY treatment sensitizes TMZ-pretreated cells to Fimepinostat. These data support the notion that the new drug combination could be effective in cells emerging after selection by TMZ treatment. Please also refer to **Topic 3** in the **Response to All Reviewers**.

Minor points

8. Please adjust the thickness of the scale bar to the same size in IF images.

Thank you for pointing out this mistake. We have adjusted the thickness of the scale bars of the IF images in the revised manuscript.

9. Figure 1F, please show the same scaled images.

We apologize for this oversight. We updated the image panel scales in **Fig. 1F**.

10. Figure EV1D, need a better quality image.

We have now included a higher resolution image of the CNV plots in **Fig. EV2D** (prev. EV1D).

11. Figure EV1F, x-y axis is missing.

We added the missing axes.

12. Figure 2C is an IF image, not "immunohistochemical staining" (in the main text).

Thank you for noticing. We have removed the statement from the manuscript text.

Coricor G, Serra R (2016) TGF-beta regulates phosphorylation and stabilization of Sox9 protein in chondrocytes through p38 and Smad dependent mechanisms. *Sci Rep* 6: 38616

Wang J, Xu SL, Duan JJ, Yi L, Guo YF, Shi Y, Li L, Yang ZY, Liao XM, Cai J *et al* (2019) Invasion of white matter tracts by glioma stem cells is regulated by a NOTCH1-SOX2 positive-feedback loop. *Nat Neurosci* 22: 91-105

Younes A, Berdeja JG, Patel MR, Flinn I, Gerecitano JF, Neelapu SS, Kelly KR, Copeland AR, Akins A, Clancy MS *et al* (2016) Safety, tolerability, and preliminary activity of CUDC-907, a first-in-class, oral, dual inhibitor of HDAC and PI3K, in patients with relapsed or refractory lymphoma or multiple myeloma: an open-label, dose-escalation, phase 1 trial. *Lancet Oncol* 17: 622-631

Reviewer 2

1. Figure 1A. The methodology used for analysis of the GBM cohort is not reported in the Methods section and needs to be added. Moreover, analysis of 'MES2' is missing from the 'Nefel signature' and should be added, even in case of not significant enrichment score.

We have included the non-significant signatures in **Fig. 1A**. The corresponding analysis method had already been included in our original submission. In the revised manuscript it can be found under the heading "Public data analysis" at the bottom of **page 12**.

2. Figure 1F-G. These results are insufficient to support the conclusion that SOX10KD prevents differentiation. Expression of differentiation markers (e.g. oligodendrocytic: GalC, astrocytic: GFAP, neuronal: beta3-tub) needs to be shown. In addition, culturing cells in 10% FBS is probably not the best condition to induce differentiation, as proliferative cues may prevail. The use of lower FBS concentrations is recommended (1-2%). Along with SOX2, SOX9 and SOX10 should be shown in the western blot, as a further validation of the analysis shown in Fig. 1E

Thank you for this suggestion. We now are presenting Western blotting analysis of Sox2, Sox9 and Sox10 protein expression in mGB1 cells in **Fig. 1G** of the revised manuscript. In addition, we cultured mGB1 cells with 2% FBS and analyzed O4, Gfap, and beta3-tubulin expression by immunofluorescence microscopy, to address Sox10-KD-induced glioblastoma cell state changes. In vitro differentiation with 2% FBS resulted in an increase of Gfap expression (**Fig. 1H**), consistent with a transition towards a more AC-like /NSC-like state. Expression of beta3-tubulin appeared to decrease but was not strongly affected. GalC, a marker of myelinating oligodendrocytes that appear late in normal lineage differentiation, was not detected. This is consistent with studies showing oligodendrocyte differentiation from tumor cells to require 2-3 weeks using 10% FBS (Gunther *et al*, 2008; Liu *et al*, 2022). In addition, it is important to emphasize that lineage-marker expression in progenitor-like cells in the tumor context in vitro might be similar but not identical to that observed in normal neurodevelopment.

In addition, we would like to highlight existing evidence supporting the dedifferentiated phenotypes of SOX10-low tumor cells. In melanoma, where SOX10 is more ubiquitously expressed, Tsoi et al. (Tsoi *et al*, 2018) demonstrated that BRAF inhibition can induce the dedifferentiation of melanoma cells from a melanocytic state (SOX10/MITF high) to an undifferentiated state (SOX10-low/SOX9 high) that is resistant to BRAF inhibition.

3. To introduce the point that GBM therapy could impinge on SOX10 expression, the cohort analyses shown in Fig. 2D-E could be presented first in this section. As SOX9 is often mutually exclusive with SOX10, it would be interesting to see the result of SOX9 analysis as well, both in the overall cohorts and in representative IHC. However, it must be stressed that the expression profile of recurrent tumors is strongly influenced not only by TMZ therapy, but also by radiotherapy, which patients almost invariably receive. Indeed, in the literature, SOX10 downregulation has been amply correlated with the effect of ionizing radiations. If selection of cases that received TMZ only is impossible, or the related statistics cannot reach significance for the low number, evidence in favor of a meaningful relationship between SOX10 expression and prior TMZ therapy remains weak, so this analysis would be better removed.

Thank you for this insightful comment. We agree that we cannot distinguish the effects of radiotherapy and TMZ chemotherapy. Therefore, we removed the corresponding analyses from the manuscript as discussed in **Topic 1** in the **Response to All Reviewers**.

4. For in vitro and in vivo experiments, the authors use multiple models including original mouse and human GBM neurospheres and organoids that are defined as 'previously characterized'. However, the main characteristics of these models should be reported again, including the essential genetic and molecular information, as they are crucial to interpreting the results, in particular those related to TMZ effect. A notable omission in the 'MGMT status' of the cells, i.e., the expression of the dealkylating enzyme MGMT, which repairs DNA damage inflicted by TMZ and is currently known as the main factor of resistance to TMZ therapy. It is recommended to assess MGMT protein expression and not simply MGMT promoter methylation as is routinely tested in patients. As the TMZ dose used in experiments such as those represented in Fig. 2B and EV3 is quite high (100 μ M) and able to effectively kill MGMT-negative GBM cells, the partial response observed in these models suggests that they may be MGMT-positive and thus inherently resistant to the drug. As a consequence, the effects observed after TMZ treatment with high doses can be unspecific, and anyway not representative of the TMZ doses used in patients. In summary: the MGMT expression of the models need to be shown and TMZ treatment need to be performed with dose curves (5-100 micromolar), and the results reevaluated in light of new evidence. In addition, the treatment protocols need to be clearly described in the Methods section (this important information is currently missing).

Thank you for pointing out the missing description of parts of the treatment protocols. We have now amended the corresponding methods on **page 12** of the revised manuscript. In addition, we confirmed the MGMT status of the used glioblastoma spheroid cell lines by Western blotting analysis as suggested by the reviewer (**Fig. EV1**). NCH644, used for the cerebral organoid experiment, was found to express MGMT at a level comparable to Jurkat cells, which we used as the positive control. NCH421k and NCH441 were MGMT-negative. These findings are consistent with the TMZ IC₅₀'s of NCH644 (227 μ M) and NCH421k (272 μ M) reported by Dirkse et al, 2019 (Dirkse *et al*, 2019).

This manuscript uses TMZ to induce SOX10 downregulation and enrich SOX9/p27-pos slow-cycling cells, showing that TMZ, which is used in glioblastoma standard therapy, can affect SOX10 expression and potentially qNSC state. The mechanism mediating SOX10 downregulation is unknown, but studies suggest it probably is not exclusively TMZ-dependent and might not be affected by MGMT methylation status (for details please refer to **Topic 1** in the **Response to All Reviewers**). This is further supported by the observation that MGMT-negative NCH421k and NCH441 cells and MGMT-positive NCH644 cells respond similarly to treatment with 25 μ M and 100 μ M TMZ, respectively. The concentration of 25 μ M was not tested for the NCH644/organoid model; however, given the IC₅₀ of NCH644 is 227 μ M (Dirkse et al, 2019), the applied 100 μ M concentration is not excessive. Thus, the MGMT response we are studying might indeed not be MGMT-status-specific but "unspecific", as the reviewer states. However, this does not invalidate its use in this study, as an inducer of SOX10 repression.

Furthermore, the effect of TMZ as a single agent or in combination with other drugs is not the focus of this manuscript. We think that the extensive experiments suggested by the reviewer would complicate the already complex study and distract the reader's attention.

5. Fig. 2A-C. In TMZ-treated cells, besides SOX10 expression, it would be interesting to show SOX9 as well.

Thank you for this comment. We agree. In the original manuscript we had shown TMZ-induced SOX9 upregulation only in Fig 2A (now **Fig. EV1A**). For the Cerebral Organoid-Glioblastoma Co-culture (GLICO) model in Fig. 2B (now Fig. EV1B), marker expression was analyzed by flow cytometry, due to limited amount of available material. SOX9 analysis was not performed since we could not find a suitable antibody for this purpose. For the analysis of the TMZ-treated RCAS tumors in Fig. 2C we now added the Sox9 immunofluorescence staining (**Fig. EV3H**).

6. Fig. 3D. In single cell analysis, the significantly higher proportion of G2/M cells observed in SOX10-KD tumors vs. controls is interesting and should be more thoroughly interpreted (page 5). Accumulation of cells in G2/M can be consistent with faster growth but also with a late cell cycle block due to detection of irreparable DNA damage (mitotic catastrophe). Or shall we postulate a role for overexpressed p27? In addition, the percentages referring to cells in G1 and S phase should be reported.

Thank you for the interesting question. We are now reporting the percentages of G1 and S-phase cells in **Fig. EV2F** and show by gamma-H2AX staining that Sox10-KD tumors do not exhibit clear evidence of DNA damage (**Fig. EV2G**). We are now also including a brief discussion of the enrichment G2/M-phase cells in Sox10-KD vs Ctrl tumors (**page 5**).

7. Fig. 3E. Surprisingly, the 'Nefel signature', originally developed in for GBM single cell analysis, and applied by these authors to analyses of GBM tissues was not applied to single cells derived to SOX10-KD. Why not? Can this analysis be added?

We initially used the human signatures of Nefel to conduct an analysis of the association between SOX10 expression and various glioblastoma molecular subtype and cell-state signatures. This analysis already indicated the association of AC-like and NSC-like cell states with low SOX10 expression status (Fig. 1A). In the further course of the analysis, we focused on the enrichment of AC-like and NSC-like signatures that also was evident in the mouse Sox10-KD tumors. We have now added the AUCell analysis of the corresponding Nefel signature in **Fig. 2F** (previously Fig. 3E).

8. Fig. 3F. Identification of NSC marker upregulation at RNA level needs to be accompanied by validation of protein expression in GBM tissues (e.g. by immunofluorescence). This could be conveniently shown in main Figure by relocating the analysis presented in Fig. 3G in EV. It is suggested to extract from Fig. 3G information related to cluster 12 and to integrate it with validation of protein expression in tumor tissues. As cluster 12 represents a tiny cell fraction, a spatial RNA analysis (RNAscope) could be conveniently added. Addition of these data (to be shown in main Figure) is required to robustly support the conclusion that SOX10-KD tumors include a fast-cycling and a slow-cycling NSC population.

To validate NSC-marker upregulation in Sox10-KD tumors, we added staining of Sox2 and Nestin proteins in KD and control tumor sections (**Fig. 2G, H**). To support our hypothesis regarding the presence of a small cell population with qNSC properties in KD tumors, we combined cell-cycle-stage analysis of the scRNA-Seq data (**Fig. EV2I**) and immunofluorescence staining of the stem-and progenitor-cell marker Sox2 and the proliferation marker Ki67 (**Fig. EV2J**). These analyses support the G0-state of the putative qNSC cluster C12, and indicated the presence of a proliferative (Ki 67+) and a non-proliferative (Ki67-) Sox2-positive population, consistent with our hypothesis.

We did not attempt validation of the scRNA-Seq data by RNA-scope analysis since we believe adding an additional technologically demanding analysis is beyond the scope of this manuscript.

9. Data presented in EV2A cannot be interpreted by this reviewer, as the related methodology is not reported in the Methods section. In the figure legend the content of Q1-Q4 scatter plot areas is not indicated.

We apologize for failing to include the methods for the analysis shown in Fig. EV2A. We added it to the revised manuscript (**page 14**). Furthermore, we removed Fig EV2A from the revised manuscript to comply with the required limitation of EV figures. However, we retained Fig. EV2B (now **Fig. EV 4A**) showing the data quantification.

The updated legend of the now removed Fig. EV2A would have read as follows:

Scatter plots showing the cell cycle distribution of mGB1 control and Sox10-KD cells (KD#1 and KD#2) after 7-day culture in stem-like or differentiating conditions and after re-supplying serum (recovered conditions). To distinguish between G0 and G1 phases, Ki67 was co-stained with DNA dye FxCycle 450. G0 (bottom left quadrant, DNA-low/Ki67-negative); G1 (top left quadrant, DANN-low/Ki67-positive); S-G2M (top right quadrant, DNA-intermediate or high/Ki67-positive)

10. As the Results paragraph starting page 6 emphasizes that 'TMZ treatment and SOX10-KD enrich a qNSC-like state in vitro', data concerning TMZ-treated cells should be presented in the main figure. However, as in the case of Fig. 2B, it is impossible to evaluate whether the model can be representative of human GBMs that are sensitive to TMZ therapy, as information about MGMT expression is missing. After the high TMZ dose used (100 μ M, again the therapeutic schedule is missing), one would expect that MGMT negative GBM cells would be completely eradicated. The presence of residual cells suggests that these cells were likely MGMT-positive at the origin, and the toxic effect of TMZ is largely unspecific. If this is the case, the model would not be representative of human treatment, where MGMT-positive cells are fully resistant to the TMZ doses (less inferior to 100 μ M) that can reach the tumor. In summary, to show that, after a TMZ treatment mimicking human treatment, a resistant qNSC population remains that could be treated by the strategy developed in SOX10-KD tumors, it would be required to utilize two independent MGMT-negative models, treated with doses of TMZ not exceeding 50 μ M (ideally 5 μ M x 3 administrations) and analyze the cell population emerging after this selection.

Please refer to **Topic 1** in the **Response to All Reviewers**, and to our answer to **comment 4** above. In summary, we do not follow the reviewer's argument that cell line MGMT status and the precise emulation of the TMZ treatment applied in human glioblastoma therapy are of critical importance to our study.

11. If the authors intend to robustly support that TMZ-treated cells mirror SOX10-KD models, they should repeat at least the in vitro experiments of LY411575-Fimepinostat therapy in combination with TMZ treatments (in MGMT- models). The authors should investigate whether the new drug combination could be effective in cells emerging after selection by TMZ treatment.

Thank you for this suggestion. We agree that adding these data would improve the manuscript. Therefore, we performed LY+Fimepinostat treatment of TMZ-pretreated and control human glioblastoma cells. The analysis, included in **Fig. EV4D** of the revised manuscript, indicates that LY treatment sensitizes TMZ-pretreated cells to Fimepinostat.

These data support the notion that the new drug combination could be effective in cells emerging after selection by TMZ treatment.

12. The single-cell analysis on experimental tumors treated with LY411575-Fimepinostat appears as a technological display failing to fulfill the promise of increasing insight into the nature of glioblastoma or its evolution under therapeutic pressure. Without showing that LY411575-Fimepinostat is a candidate therapy for glioblastoma, to investigate the profile of single cells treated by this therapy is of dubious significance. The message that the relapsing tumor tend to display a mesenchymal profile, similar to tumor relapsing after conventional therapy should be further elaborated or, perhaps, better served as a starting point in future work.

We disagree with the reviewer's opinion that using Fimepinostat for analyzing SOX10-dependent cell-state plasticity is moot before its validation as a candidate for glioblastoma therapy (which also would have to be done in an independent study). Rather, we think our data establish a novel mechanism of SOX10 affecting glioblastoma aggressiveness and overall survival and provide a rationale for designing tumor therapies based on single-cell analysis. However, we thank the reviewer for pointing out that we overemphasized the more mesenchymal character of the qNSC clusters emerging after treatment. The important message here is that qNSC-like clusters re-emerge. Their mesenchymalness is less important. We adapted the text accordingly (**page 6**).

Minor points

N.1. The nomenclature for the p53 gene should be updated from 'Trp53' to 'Tp53'.

We updated the gene name to Tp53.

N.2. Fig. 1D. The use of EGFR and ERBB3 as markers of SOX9 and SOX10 positive cells (NSC and OLC, respectively) should be more extensively explained. In Figure 1 legend the acronym NLC should be corrected in 'NSC'.

We corrected the acronym NLC to NSC; thank you for noticing. In addition, we added more information on the NSC and OLC marker genes to the results (**page 4**).

N.3. Page 7. Fig. EV4C seems wrongly cited.

We corrected the corresponding citation, legend and figure panel.

Dirkse A, Golebiewska A, Buder T, Nazarov PV, Muller A, Poovathingal S, Brons NHC, Leite S, Sauvageot N, Sarkisjan D *et al* (2019) Stem cell-associated heterogeneity in Glioblastoma results from intrinsic tumor plasticity shaped by the microenvironment. *Nat Commun* 10: 1787

Gunther HS, Schmidt NO, Phillips HS, Kemming D, Kharbanda S, Soriano R, Modrusan Z, Meissner H, Westphal M, Lamszus K (2008) Glioblastoma-derived stem cell-enriched cultures form distinct subgroups according to molecular and phenotypic criteria. *Oncogene* 27: 2897-2909

Liu J, Wang X, Chen AT, Gao X, Himes BT, Zhang H, Chen Z, Wang J, Sheu WC, Deng G *et al* (2022) ZNF117 regulates glioblastoma stem cell differentiation towards oligodendroglial lineage. *Nat Commun* 13: 2196

Tsoi J, Robert L, Paraiso K, Galvan C, Sheu KM, Lay J, Wong DJL, Atefi M, Shirazi R, Wang X *et al* (2018) Multi-stage Differentiation Defines Melanoma Subtypes with Differential Vulnerability to Drug-Induced Iron-Dependent Oxidative Stress. *Cancer Cell* 33: 890-904 e895

Reviewer 3

1. In Figure 1E, it would be helpful to clarify whether the GFP labels tumor cells. If so, it appears that the SOX9 staining is observed in GFP-negative stromal cells, which does not support their conclusion that SOX10 KD induce SOX9 expression in tumor cells.

Sox9 is also expressed in normal astrocytes and, therefore, detected in some GFP-negative stroma cells. The antagonistic relationship between SOX10 and SOX9 has been well-described in both normal and tumor cell settings. In normal glial cell development SOX10 upregulates mir-335 and mir-338 during late OPCs to inhibit the expression of the SOX9, an essential regulator for the neural progenitor cell state, to prevent de-differentiation (Reiprich *et al*, 2017). The antagonist relationship of SOX9 and SOX10 is preserved in glioblastoma (Wang *et al*, 2020), and SOX10 expression is decreasing and that of SOX9 increasing along a de-differentiation and malignant progression axis in melanoma (Shakhova *et al*, 2015; Tsoi *et al*, 2018).

2. In Figure 1G, the authors should include the immunoblotting data for SOX10 and SOX9.

Thank you for pointing out this oversight. We now added the Sox9 and Sox10 Western blotting data to **Fig. 1G**.

3. Figure 2C should address whether Sox9 expression is upregulated upon TMZ treatment.

SOX9 upregulation upon TMZ treatment was shown in Figure EV3D of the original submission. This corresponds to **Fig. EV3H** of the revised manuscript.

4. For Figure 2E, the authors could strengthen the analysis to perform SOX10 staining in multiple samples and provide quantification. A scale bar should be included.

We have excluded this analysis from the revised manuscript since we cannot separate the effects on SOX10 expression of TMZ and radiation therapy in patient tumor therapy (please see **Topic 1** in the **Response to All Reviewers**).

5. In Figures 2D and 2E, the authors should also compare the expression levels of SOX9 between primary and recurrent tumors.

The analyses of therapy-induced effects on SOX10 expression in patient tumors are no longer part of this manuscript since we cannot separate the effects on SOX10 expression of TMZ and radiation therapy in patient tumor therapy (please also see **Topic 1** in the **Response to All Reviewers**).

6. In Figure 3G, why cluster 10 was removed? Additionally, clusters 4, 3, and 11, labeled as differentiated, should be further categorized into specific neural lineage differentiations, such as astrocytes, neurons, or oligodendrocytes.

We are very sorry for this mistake. There was no reason to exclude cluster 10. It was accidentally removed. We have now re-included cluster 10 in the revised figure **Fig. 2E** (prev. Fig. 3G). It is important to emphasize that lineage-marker expression in progenitor-like cells in the tumor context in vitro might be similar but not identical to that observed in normal neurodevelopment. Thus, the unambiguous assignment of tumor cell clusters to normal developmental lineages is challenging and potentially misleading. We do not think it essential

for this manuscript. But we are now providing the DEG lists of all clusters in the **Appendix** so that the readers can form their own opinion.

7. In Figure 5F and Figure 6, to determine the synergistic effects of LY and Fimepinostat, it is essential for the authors to incorporate four distinct treatment groups: control, individual drug treatments (LY or Fimepinostat), and combination therapy (LY and Fimepinostat) for both in vitro and in vivo experiments. Moreover, in Figure 5F, they should perform the same experiment in mGB1-shRNA-control cells.

We added survival, and Ki67 staining data of tumors of Fimepinostat-treated animals to the revised manuscript. These data do not allow to differentiate synergistic from additive effects, but they do indicate a survival advantage of the combination-treatment group over the Fimepinostat treatment group (**Fig. EV5**). In addition, we confirmed the synergy of LY and Fimepinostat, which we had observed in mGB1 Sox10-KD cells (Fig. 5G), in TMZ-pretreated human NCH421k and NCH441 cells (**Fig. EV4D**).

Finally, we want to emphasize that identifying drug synergisms between Fimepinostat and LY and TMZ was not the focus of our study. Please also refer to **Topic 3** in the **Response to All Reviewers**.

8. In Figure 6I, combo treatment induced MES-like signature. It is known that the MES-like signature is associated with immune response. Did the author observe alterations in the immune microenvironment between control and combo groups?

Thank you for the interesting question. Yes indeed, using Iba1 immuno-histochemical staining, we were able to detect increased numbers of tumor-associated macrophages in the tumors of the combination-treatment group relative to the control group. This finding points to potentially interconnected therapy-induced effects in the tumor and immune-cell compartments that should be investigated in future studies.

Figure 3. Immunohistochemistry staining of Iba1 (a tumor-associated myeloid cell marker) in Vehicle (upper) and Combo (bottom) group. tumors. Two fields of view were shown. Scale bars = 50 μ m

9. Finally, elucidating the mechanism underlying SOX10's regulation of GBM stemness is imperative to strengthen the overall conclusion of this manuscript.

SOX10 is a lineage-differentiation factor that acts antagonistically to SOX9, an essential regulator for the neural progenitor cell state (Varn *et al*, 2022; Wang *et al.*, 2020). The here observed cell de-differentiation after SOX10 removal is consistent with the literature; e.g., GBM stemness in Sox10-KD tumors might be not only be induced by diminished Sox10-dependent differentiation cues, but also the activation of a SOX9-SOX2-Notch positive feedback loop (Wang *et al*, 2019), consistent with our findings (discussion, **page 9**). We think that the analysis of further mechanistic details is beyond the scope of this study.

Reiprich S, Cantone M, Weider M, Baroti T, Wittstatt J, Schmitt C, Kuspert M, Vera J, Wegner M (2017) Transcription factor Sox10 regulates oligodendroglial Sox9 levels via microRNAs. *Glia* 65: 1089-1102

Shakhova O, Cheng P, Mishra PJ, Zingg D, Schaefer SM, Debbache J, Hausel J, Matter C, Guo T, Davis S *et al* (2015) Antagonistic cross-regulation between Sox9 and Sox10 controls an anti-tumorigenic program in melanoma. *PLoS Genet* 11: e1004877

Tsoi J, Robert L, Paraiso K, Galvan C, Sheu KM, Lay J, Wong DJL, Atefi M, Shirazi R, Wang X *et al* (2018) Multi-stage Differentiation Defines Melanoma Subtypes with Differential Vulnerability to Drug-Induced Iron-Dependent Oxidative Stress. *Cancer Cell* 33: 890-904 e895

Varn FS, Johnson KC, Martinek J, Huse JT, Nasrallah MP, Wesseling P, Cooper LAD, Malta TM, Wade TE, Sabedot TS *et al* (2022) Glioma progression is shaped by genetic evolution and microenvironment interactions. *Cell* 185: 2184-2199 e2116

Wang J, Xu SL, Duan JJ, Yi L, Guo YF, Shi Y, Li L, Yang ZY, Liao XM, Cai J *et al* (2019) Invasion of white matter tracts by glioma stem cells is regulated by a NOTCH1-SOX2 positive-feedback loop. *Nat Neurosci* 22: 91-105

Wang Z, Sun D, Chen YJ, Xie X, Shi Y, Tabar V, Brennan CW, Bale TA, Jayewickreme CD, Laks DR *et al* (2020) Cell Lineage-Based Stratification for Glioblastoma. *Cancer Cell* 38: 366-379 e368

Dear Bernhard,

Thank you for the submission of your revised manuscript. We have now received the enclosed reports from the referees and I am happy to say that all of them support the publication of your work now. Referee 2 has a few more minor suggestions that I would like you to incorporate before we can proceed with the official acceptance of your manuscript. Please co-submit a point-by-point response to all final requests.

A few editorial requests will also need to be addressed:

- Your ms has 5 main figures, but the results and discussion sections are not combined. Please either add one more main figure to publish your work as a full article, or combine the results and discussion sections to publish it as a short report (with a maximum of 29,000 characters excluding references and materials and methods).
- The Data Availability Section needs to be moved to before the Acknowledgments.
- The conflict of interest subheading needs to be corrected to "Disclosure Statement and Competing Interests"
- The corresponding author(s) need(s) to be clearly marked on the title page, and their email address should be provided on the title page.
- AC/CRedit: need to be removed from the ms file. All credits need to be entered during ms submission.
- The author CHECKLIST is missing responses from the pull-down menus in D87 and D88. Please send us a fully completed checklist. Also all questions in the statistics section need to be answered.
- All FUNDING INFO needs to be entered also in our online ms submission system. All acknowledged funders from the ms need to be inserted as separate Funder entries.
- There is a mismatch between the tables names in the Excel sheet and the actual EV table. Information in Tables EV1-EV4 should be moved to the Reagents and Tools table that we would like to ask you to upload. You can download a word template from our guide to authors under "structured methods". Tables EV5-EV20 are actual Datasets, as far as I understand, and can be combined in one file called Dataset EV1 with several tabs, if these are related data. Such a Dataset EV1 file needs a first tab with a title or legend that explains what each tab shows. Alternatively, you can upload several Datasets EV1-EVx (there is no limit in the total number). All ms callouts need to be updated accordingly.
- Materials & Methods should be called "Methods".
- In Figure 5D and Figure EV5C, it appears that images are re-used in both figures. Figure EV5 legend says - related to Figure 5. This should be extended and the re-use of cells/images should be included in the Figure EV5 legend.
- Please note that the PRJEB77072 dataset needs to be freely accessible upon the online publication of your ms. Also, the specific URL for the PRJEB77072 dataset needs to be provided in the data availability statement.
- Please define the annotated p values ***/** as well as provide the exact p-values for the same in the legend of figure EV 1b; as appropriate.
- Please note that the exact p values are not provided in the legends of figures 1b-c; 2f; 3e, h; 4b, d; 5e, k; EV 2f, h; EV 3f-g; EV 5d.
- Please indicate the statistical test used for data analysis in the legends of figures 1c-d; 3a-b; EV 2h; EV 3f.
- Please note that the box plots need to be defined in terms of minima, maxima, centre, bounds of box and whiskers, and percentile in the legends of figures 2f; 4d; 5k; EV 2f.
- Please note that the box plots need to be defined in terms of minima, maxima, bounds of box and whiskers in the legends of figures 3e; EV 4c.
- Please note that information related to n is missing in the legends of figures 2f; 4b; 5k; EV 2f; EV 5b.
- Although 'n' is provided, please describe the nature of entity for 'n' in the legends of figures EV 3c; EV 4a.

- Please note that the error bars are not defined in the legends of figures 4b; EV 5b.

I would like to suggest some minor changes to the abstract that needs to be written in present tense. Do you agree with:

Phenotypic plasticity is a cause of glioblastoma therapy failure. We previously showed that suppressing the oligodendrocyte-lineage regulator SOX10 promotes glioblastoma progression. Here, we analyze SOX10-mediated phenotypic plasticity and exploit it for glioblastoma therapy design. We show that low SOX10 expression is linked to neural-stem-cell (NSC)-like glioblastoma cell states and is a consequence of temozolomide treatment in animal and cell line models. Single-cell transcriptome profiling of Sox10-KD tumors indicate that Sox10 suppression is sufficient to induce tumor progression to an aggressive NSC/developmental-like phenotype, including a quiescent NSC-like cell population. The quiescent NSC state is induced by temozolomide and Sox10-KD and reduced by Notch pathway inhibition in cell line models. Combination treatment using Notch and HDAC/PI3K inhibitors extends the survival of mice carrying Sox10-KD tumors, validating our experimental therapy approach. In summary, SOX10 suppression mediates glioblastoma progression through NSC/developmental cell state transition, including the induction of a targetable quiescent-NSC state. This work provides a rationale for the design of tumor therapies based on single-cell phenotypic plasticity analysis.

EMBO press papers are accompanied online by A) a short (1-2 sentences) summary of the findings and their significance, B) 2-3 bullet points highlighting key results and C) a synopsis image that is exactly 550 pixels wide and 200-600 pixels high (the height is variable). The synopsis image should provide a sketch of the major findings, like a graphical abstract. Please note that text needs to be readable at the final size. Please send us this information along with the final manuscript.

Referee #1:

I'm satisfied with the revised version.

Referee #2:

The authors have addressed my concerns by conducting additional experiments, clarifying their perspective in their Reply, and revising the text to downplay conclusions that were not sufficiently supported by the data. However, before I can fully endorse the manuscript for publication, I believe the following points need further attention.

Point #1, concerning Points #4 and #10 of my previous review:

The authors have performed the suggested analyses and experiments, leading them to conclude that the effect of temozolomide on SOX10 expression-observed in both MGMT-expressing and MGMT-non-expressing models-is likely independent of the cell's ability to repair the DNA damage caused by temozolomide. This is a significant finding that should be more thoroughly discussed and highlighted in the Abstract. Although the authors aim to present an innovative approach to glioblastoma treatment, they should not overlook the fact that real-world progress for patients will inevitably involve clinical trials that include the concomitant or sequential administration of temozolomide, along with the stratification of patients based on MGMT status (MGMT+ or MGMT-). In summary, the authors should be more concerned that their study is of critical importance for glioblastoma therapy, rather than the 'MGMT status and the precise TMZ treatment applied in human glioblastoma therapy are of critical importance to their study'.

Additionally, the previously measured TMZ IC50 values in the models should be reported in the text.

Point #2, concerning Point #11 of my previous review:

The addition of the experiment involving combination treatment with Fimepinostat and LY in temozolomide-pretreated models (new Fig. EV4D) is appreciated. However, the results do not achieve statistical significance, and this should be explicitly stated in the Results section. Furthermore, the conditions of temozolomide pretreatment must be described in the Methods section.

Referee #3:

The authors have addressed my comments with supporting data or excellent explanation. No further comment.

Point-by-point response to reviewers

Many thanks to the reviewers and the editorial team for their suggestions and continued support. We highlighted all edits in red type in the revised manuscript.

Referee #1:

I'm satisfied with the revised version.

Thank you for helping us improve the manuscript.

Referee #2:

The authors have addressed my concerns by conducting additional experiments, clarifying their perspective in their Reply, and revising the text to downplay conclusions that were not sufficiently supported by the data. However, before I can fully endorse the manuscript for publication, I believe the following points need further attention.

Point #1, concerning Points #4 and #10 of my previous review:

The authors have performed the suggested analyses and experiments, leading them to conclude that the effect of temozolomide on SOX10 expression-observed in both MGMT-expressing and MGMT-non-expressing models-is likely independent of the cell's ability to repair the DNA damage caused by temozolomide. This is a significant finding that should be more thoroughly discussed and highlighted in the Abstract. Although the authors aim to present an innovative approach to glioblastoma treatment, they should not overlook the fact that real-world progress for patients will inevitably involve clinical trials that include the concomitant or sequential administration of temozolomide, along with the stratification of patients based on MGMT status (MGMT+ or MGMT-). In summary, the authors should be more concerned that their study is of critical importance for glioblastoma therapy, rather than the 'MGMT status and the precise TMZ treatment applied in human glioblastoma therapy are of critical importance to their study'.

We sincerely thank you for your helpful and stimulating comments, current and previous. Thank you for helping us improve the manuscript.

We are now explicitly stating that SOX10 downregulation by TMZ does not appear to depend on MGMT status (page 4). By explicitly reporting this information in the Results section, we intend to make it better accessible to the readers. We did not include it in the abstract, however, since the suppression of SOX10 by different types of therapy is not our finding, but first was demonstrated in the studies cited at the start of the paragraph.

Additionally, the previously measured TMZ IC50 values in the models should be reported in the text. We added the TMZ IC50 values of the glioblastoma cell lines on page 4, second paragraph.

Point #2, concerning Point #11 of my previous review:

The addition of the experiment involving combination treatment with Fimepinostat and LY in

temozolomide-pretreated models (new Fig. EV4D) is appreciated. However, the results do not achieve statistical significance, and this should be explicitly stated in the Results section.
We added a corresponding explicit statement at the bottom of page 7.

Furthermore, the conditions of temozolomide pretreatment must be described in the Methods section.
Thank you for noticing. We added the information to the methods section (page 12).

Referee #3:

The authors have addressed my comments with supporting data or excellent explanation. No further comment.
Thank you for helping us improve the manuscript.

Point-by-point response to the editorial requests

- Your ms has 5 main figures, but the results and discussion sections are not combined. Please either add one more main figure to publish your work as a full article, or combine the results and discussion sections to publish it as a short report (with a maximum of 29,000 characters excluding references and materials and methods).

We split the very busy Fig. 5 into the new Figs. 5 and 6, separating the survival analysis (Fig. 5A-E) and the scRNA-seq analysis of the treated tumors (prev. Fig. 5F-K; now Fig. 6A-F).

- The Data Availability Section needs to be moved to before the Acknowledgments.

Done

- The conflict of interest subheading needs to be corrected to "Disclosure Statement and Competing Interests"

Done

- The corresponding author(s) need(s) to be clearly marked on the title page, and their email address should be provided on the title page.

Done

- AC/CRedit: need to be removed from the ms file. All credits need to be entered during ms submission.

Done

- The author CHECKLIST is missing responses from the pull-down menus in D87 and D88. Please send us a fully completed checklist. Also all questions in the statistics section need to be answered.

Done

- All FUNDING INFO needs to be entered also in our online ms submission system. All acknowledged funders from the ms need to be inserted as separate Funder entries.

Done

- There is a mismatch between the tables names in the Excel sheet and the actual EV table. Information in Tables EV1-EV4 should be moved to the Reagents and Tools table that we would like to ask you to upload. You can download a word template from our guide to authors under "structured methods".

Tables EV5-EV20 are actual Datasets, as far as I understand, and can be combined in one file called Dataset EV1 with several tabs, if these are related data. Such a Dataset EV1 file needs a first tab with a title or legend that explains what each tab shows. Alternatively, you can upload several Datasets EV1-EVx (there is no limit in the total number). All ms callouts need to be updated accordingly.

We followed your suggestion and distributed the tables between a Regents and Tools table and Dataset EV1.

- Materials & Methods should be called "Methods".

Done

- In Figure 5D and Figure EV5C, it appears that images are re-used in both figures. Figure EV5 legend says - related to Figure 5. This should be extended and the re-use of cells/images should be included in the Figure EV5 legend.

Done

- Please note that the PRJEB77072 dataset needs to be freely accessible upon the online publication of your ms. Also, the specific URL for the PRJEB77072 dataset needs to be provided in the data availability statement.

We released the dataset on 26.08.2024. It should be searchable in the ENA database within 48 hours.

- Please define the annotated p values ***/** as well as provide the exact p-values for the same in the legend of figure EV 1b; as appropriate.

The exact P-value was added to the Figure. The * notation was removed.

- Please note that the exact p values are not provided in the legends of figures 1b-c; 2f; 3e, h; 4b, d; 5e, k; EV 2f, h; EV 3f-g; EV 5d.

We are now providing the exact P values in the figures. We removed all “p<xx” and * notations.

- Please indicate the statistical test used for data analysis in the legends of figures 1c-d; 3a-b; EV 2h; EV 3f.

We indicated the statistical tests used in the respective figure legends.

- Please note that the box plots need to be defined in terms of minima, maxima, centre, bounds of box and whiskers, and percentile in the legends of figures 2f; 4d; 5k; EV 2f.

We made the suggested changes in the respective figure legends.

- Please note that the box plots need to be defined in terms of minima, maxima, bounds of box and whiskers in the legends of figures 3e; EV 4c.

We made the suggested changes in the respective figure legends.

- Please note that information related to n is missing in the legends of figures 2f; 4b; 5k; EV 2f; EV 5b.

We added the missing information to the respective figure legends.

- Although 'n' is provided, please describe the nature of entity for 'n' in the legends of figures EV 3c; EV 4a.

These are biological replicates. The information now is provided.

- Please note that the error bars are not defined in the legends of figures 4b; EV 5b.

The information (Fig. 4B, mean SEM; Fig. EV5B, mean SD) now is provided in the legends.

I would like to suggest some minor changes to the abstract that needs to be written in present tense. Do you agree with:

Phenotypic plasticity is a cause of glioblastoma therapy failure. We previously showed that suppressing the oligodendrocyte-lineage regulator SOX10 promotes glioblastoma progression. Here, we analyze SOX10-mediated phenotypic plasticity and exploit it for glioblastoma therapy design. We show that low SOX10 expression is linked to neural-stem-cell (NSC)-like glioblastoma cell states and is a consequence of temozolomide treatment in animal and cell line models. Single-cell transcriptome profiling of Sox10-KD tumors indicate that Sox10 suppression is sufficient to induce tumor progression to an aggressive NSC/developmental-like phenotype, including a quiescent NSC-like cell population. The quiescent NSC state is induced by temozolomide and Sox10-KD and reduced by Notch pathway inhibition in cell line models. Combination treatment using Notch and HDAC/PI3K inhibitors extends the survival of mice carrying Sox10-KD tumors, validating our experimental therapy approach. In summary, SOX10 suppression mediates glioblastoma progression through NSC/developmental cell state transition, including the induction of a targetable quiescent-NSC state. This work provides a rationale for the design of tumor therapies based on single-cell phenotypic plasticity analysis.

We agree. We applied all your edits to the abstract.

EMBO press papers are accompanied online by A) a short (1-2 sentences) summary of the findings and their significance, B) 2-3 bullet points highlighting key results and C) a synopsis image that is exactly 550 pixels wide and 200-600 pixels high (the height is variable). The synopsis image should provide a sketch of the major findings, like a graphical abstract. Please note that text needs to be readable at the final size. Please send us this information along with the final manuscript.

We will send this information along with the final manuscript.

Dr. Bernhard Radlwimmer
German Cancer Research Center (DKFZ)
Division of Molecular Genetics
Im Neuenheimer Feld 580
Heidelberg 69120
Germany

Dear Bernhard,

I am very pleased to accept your manuscript for publication in the next available issue of EMBO reports. Thank you for your contribution to our journal.
